

# A New Map of the Permafrost Distribution on the Tibetan Plateau

Defu Zou[1,2], Lin Zhao[1], Yu Sheng[2], Ji Chen[2], Guojie Hu[1], Tonghua Wu[1], Jichun Wu[2], Changwei Xie[1], Xiaodong Wu[1], Qiangqiang Pang[1], Wu Wang[1], Erji Du[1], Wangping Li[1], Guangyue Liu[1], Jing Li[2], Yanhui Qin[1], Yongping Qiao[1], Zhiwei Wang[1], Jianzong Shi[1] and Guodong Cheng[2]

[1]Cryosphere Research Station on Qinghai–Xizang Plateau, State Key Laboratory of Cryospheric Science, Northwest Institute of Eco–Environment and Resources, Chinese Academy of Sciences (CAS), Lanzhou, 730000, China
[2]State Key Laboratory of Frozen Soil Engineering, Northwest Institute of Eco–Environment and Resources, CAS, Lanzhou, 730000, China

*Correspondence to*: L. Zhao (linzhao@lzb.ac.cn)

**Abstract.** The Tibetan Plateau (TP) possesses the largest areas of permafrost terrain in the mid- and low-latitude regions of the world. A detailed database of the distribution and characteristics of permafrost is crucial for engineering planning, water resource management, ecosystem protection, climate modelling, and carbon cycle research. Although some permafrost distribution maps have been compiled in previous studies and have been proven to be very useful, due to the limited data source, ambiguous criteria, little validation, and the deficiency of high-quality spatial datasets, there is high uncertainty in the mapping of the permafrost distribution on the TP. In this paper, a new permafrost map was generated mostly based on freezing and thawing indices from modified Moderate Resolution Imaging Spectroradiometer (MODIS) land surface temperatures (LSTs) and validated by various ground-based datasets. The soil thermal properties of five soil types across the TP were estimated according to an empirical equation and in situ observed soil properties (moisture content and bulk density). The Temperature at the Top of Permafrost (TTOP) model was applied to simulate the permafrost distribution. The results show that permafrost, seasonally frozen ground, and unfrozen ground covered areas of $1.06 \times 10^6$ km$^2$ (40%), $1.46 \times 10^6$ km$^2$ (56%), and $0.03 \times 10^6$ km$^2$ (1%), respectively, excluding glaciers and lakes. The ground-based observations of the permafrost distribution across the five investigated regions (IRs, located in the transition zones of the permafrost and seasonally frozen ground) and three highway transects (across the entire permafrost regions from north to south) have been used to validate the model. The result of the validation shows that the kappa coefficient varies from 0.38 to 0.78 with an average of 0.57 at the five IRs and 0.62 to 0.74 with an average of 0.68 within the three transects. Compared with two maps compiled in 1996 and 2006 (kappa coefficients in average 0.06 and 0.35 in five IRs, 0.34 and 0.50 within three transects, respectively), the result of the TTOP modelling shows greater accuracy, especially in identifying thawing regions. Overall, the results provide much more detailed maps of the permafrost distribution and could be a promising basic data set for further research on permafrost on the Tibetan Plateau.




# 1 Introduction

As a main component of the cryosphere, permafrost is sensitive to climate changes (Wu et al., 2002b; Haeberli and Hohmann, 2008; Li et al., 2008; Gruber, 2012). Due to its unique and extremely high altitude with an average elevation over 4000 m (Qiu, 2008), the Tibetan Plateau (TP) (Zhang et al., 2002 and 2014) possesses the largest areas of permafrost in the mid- and low-latitude regions of the world (Zhao et al., 2004 and 2010). The presence of permafrost and its dynamics complicate the water and energy exchange between soil and atmosphere and thereby introduce greater uncertainty into Global Climate Models (GCMs) when predicting climate change (Romanovsky et al., 2002; Smith and Riseborough, 2002; Cheng and Wu, 2007; Riseborough et al., 2008; Zhao et al., 2010). To generate correct quantitative simulations, an accurate permafrost distribution of TP is evidently needed to improve the permafrost module description. Moreover, an accurate contemporary permafrost distribution map would serve as a standard to estimate permafrost degradation and as a basis for further quantitative research. Therefore, understanding the current permafrost situation on TP has become particularly urgent.

Over the past half-century, a significant amount of research has been conducted on TP permafrost distribution, and many permafrost maps (Shi and Mi, 1988; Li and Cheng, 1996; Brown et al., 1997 and 1998; Qiu et al., 2000; Wang et al., 2006) have been compiled to evaluate the distribution and thermal states of permafrost. These maps have been utilized widely to study the responses and feedback of permafrost to climate change (Ran et al., 2012). However, there is great variation in the permafrost areas and boundaries of these maps, from $1.12 \times 10^6$ to $1.50 \times 10^6$ km$^2$, due to different data collection periods, data sets, and methods (Yang et al., 2010; Ran et al., 2012). These maps represent different assessments of the permafrost distribution on the TP at different times. This spatial heterogeneity is associated with temporal heterogeneity, which leads to great difference in delineating the permafrost boundaries.

In 1980s and 90s, permafrost maps were compiled with conventional cartographic techniques that use the topographic maps as base maps, and permafrost boundaries were plotted on the base map by hand (Tong and Li, 1983; Shi and Mi, 1988; Li and Cheng, 1996). The representative and most widely used benchmark map is the *Map of Permafrost on the Qinghai-Tibetan Plateau* (Li and Cheng, 1996), whose boundary was determined mainly based on the air temperature isotherms combined with field data, satellite images and many relevant maps. After 2000, GIS software began to be applied to the mapping of permafrost. Some simple empirical models with a minimal data requirement were established to consider the permafrost characteristics on the TP, such as the elevation model (Li and Cheng, 1999) and Mean Annual Ground Temperature (MAGT) (Nan et al., 2002). Unfortunately, due to the stability of elevation and scarcity of consecutive ground temperature observations, such models are frequently insufficient in capturing regional variations of permafrost (Chen et al., 2015). Meanwhile, some models with simplified physical processes applicable to high latitude permafrost were transferred to simulate the permafrost distribution on TP such as the frost index (Nelson and Outcalt, 1983) and the Temperature at the Top of Permafrost (TTOP) (Smith and Riseborough, 1996; Wu et al., 2002a). These models link permafrost temperature with surface temperature through seasonal surface transfer functions and subsurface thermal properties, which can provide reasonable assessments of permafrost distribution when the permafrost upper





boundary conditions and regional soil thermal properties were satisfied. Most temperature fields that have previously been used in these models were also generated from spatially interpolated air temperature (Pang et al., 2011) or coarser resolution (e.g., 0.125°×0.125°) atmospheric reanalysis data (e.g., ERA-Interim) (Qin et al., 2015). Although air temperature produces inaccurate and excessively low resolution estimates of land surface temperature (LST), it was still widely used in the monitoring of permafrost in practical applications because of limited LST observations. In these

studies, the N factor has been the optimal and effective way to transform the air temperature to the LST (Klene et al., 2001; Lunardini, 1978). With the recent development of infrared remote sensing technology, an increasing number of LST products derived from different satellite images have been applied to global and regional permafrost distribution research (Hachem et al., 2009; Kääb, 2008; Langer et al., 2010; Nguyen et al., 2009; Westermann et al., 2012 and 2015). The high spatiotemporal resolution of these products should make them a preferable alternative to temperature

interpolation or reanalysis datasets (Zhang et al., 2004). However, the LSTs observed by satellite sensors are instantaneous values at passing times and must be transformed into mean daily temperature to serve as the thermal state of each day before being utilized. Wang et al. (2011) averaged the twice-daily LST observations of the Moderate Resolution Imaging Spectroradiometer (MODIS) sensors on board Terra satellite to drive the TTOP model, and their results show that there was a systematic bias with the ground observations because of different observation times (Wang

et al., 2011). More accurate calculation and validation for the remote sensing LST dataset are needed before application. In addition, the limited available soil thermal property spatial datasets create another problem when modelling permafrost distribution. Most previous soil surveys were carried out in seasonally frozen ground or permafrost along the Qinghai– Tibet Highway (Li et al., 2014; Li et al., 2015a) rather than permafrost regions in the plateau hinterland due to the harsh climate and inconvenient access. Therefore, soil thermal properties have generally been estimated via soil types

generated from a limited number of plateau geologic classification studies (Li et al., 2015b; Wang et al., 2011). Overall, there are insufficient field investigations to take part in the modelling and validate the maps and their accuracy.

Recently, plenty of field survey datasets have been obtained based on the project "Investigation of Permafrost and Its Environment over The Qinghai–Xizang(Tibet) Plateau" conducted by the Cryosphere Research Station on Qinghai– Xizang Plateau, Chinese Academy of Sciences (CAS), which could provide perfect validation data of permafrost

distribution maps. In addition, some new progress in research on remote sensing LST applications and spatial soil characteristics on the TP were studied. An empirical model of daily mean LST was established and performed well in continuous permafrost regions of the Central TP (Zou et al., 2014). Li et al. (2015b) studied the relationships between environmental factors and soil types in the permafrost region on TP, and they utilized a decision-making tree to spatialize the soil types. The results exhibited good reliability and could be used to realize the spatialization of soil thermal

properties. Based on these studies, the drive data for permafrost distribution models can likely be provided.

This study aims to generate a new permafrost distribution map on the TP combined with remote sensing LST products and the investigated soil thermal properties, and to validate the accuracy of the results in this study and the two most widely used permafrost maps. In this study, a multiple linear regression model based on MODIS LST was established, and ground-based LST observations were employed to calibrate the results. Soil thermal conductivities of each soil type



on the TP were calculated via in situ observed soil moisture content and bulk density. TTOP model was employed to simulate the permafrost distribution, and the results were validated by the observed permafrost distributions of boreholes, five investigated regions (IRs, located in the transition zones of the permafrost and seasonally frozen ground) and three transects (across the entire permafrost regions from north to south). The TTOP modelling result was also compared with that of two recent benchmark maps (made in 1996 and 2006).

## 2 Materials and methods

### 2.1 Field survey datasets

The comprehensive investigation of permafrost and its environments on the TP was conducted from 2009 to 2014. Five investigated regions (IRs)—WenQuan (WQ) (Zhang et al., 2011 and 2012) and Budongquan-Qingshuihe (B-Q) in the Eastern TP, AErJin (AEJ) in the Northeastern TP, GaiZe (GZ) (Chen et al., 2016) in the Southern TP, and XiKunLun (XKL) (Li et al., 2012) in the Western TP (Fig.1), which are located in the transition zones between permafrost and seasonally frozen ground with different climatic and geographic conditions—were selected for detailed investigation. Ground-based observations, mechanical excavation, geophysical exploration (Ground Penetrating Radar, GPR; Time-domain ElectroMagnetic, TEM), and drilling method were employed, and comprehensive surveys of the permafrost distribution boundary, soil, vegetation, climate, and landform were carried out in all five IRs. The datasets of ground temperature profiles, spatial distribution of vegetation (Wang et al., 2016) and soil types (Li et al., 2014 and 2015a) were obtained and a long-term permafrost monitoring network was established, including automatic weather station and borehole records.

### 2.1.1 Boreholes and soil pits

Field survey datasets including ground temperature, soil moisture content, and bulk density were obtained in the investigation. The ground temperature, measured by temperature probes at different depths (generally set at 0.5 m intervals from 0 to 5 m, 1 m from 5 to 20 m, 2 m from 20 to 40 m, 5 m from 40 to 60 m, and 10 m greater than 60 m) in boreholes were used to determine whether permafrost exists. The soil samples were collected according to depth increments at each pit. The field bulk density (weight of the soil per unit volume) was measured by the clod method. Samples for moisture determination were stored in aluminium sampling boxes and carefully sealed to prevent changes of soil moisture. The soil moisture content was expressed by weight as the ratio of the mass of water present to the dry weight of the soil sample (Wu et al., 2016). Considering that the sampling period was concentrated from July to October, the weighted average moisture content by depth was used to denote the mean state of each pit. The soil moisture content and bulk density were used to calculate the soil thermal conductivity. The statistics of the field survey samples show a total of 125 boreholes and 199 soil pits in five IRs (Table 1).

### 2.1.2 Permafrost maps of five investigated regions

The permafrost maps of five IRs were used as the validation data in this study. In the local region, the elevation and





terrain factors have greater influence on permafrost occurrence than that of longitude and latitude, especially on mountainous permafrost (Riseborough et al., 2008). The lower limits of permafrost (LLPs) distribution through transects located in different geographical, geomorphological conditions in each IR were determined by the geophysical investigation methods (GPR and TEM) firstly, and then validated by boreholes and pits investigation datasets (Zhang et al., 2012; Chen et al., 2016). The linear regressions between the MAGT (ground temperatures at the depth of 10 to 15 m on the TP generally) of boreholes and their elevation were analyzed, and where the elevations as MAGT equal to 0 ℃ were considered as the mean LLPs for each IR (Li et al., 2012; Chen et al., 2016). The results showed that the LLPs on the south-facing slopes are about 100-200 m higher than that on the north-facing slopes in five IRs. The permafrost map was generated for each IR based on the criteria of LLPs in different conditions combined the digital elevation model (DEM) data, and a portion of the observed results of geophysical methods and boreholes was reserved to validate the maps (Zhang et al., 2012; Li et al., 2012; Chen et al., 2016). The permafrost map of WenQuan IR was validated with all available LLPs datasets that shows an accuracy of 85.6% (Zhang et al., 2011).

### 2.1.3 Permafrost distribution of three highway transects

The three highway transects were set as follows (Fig.1): National Highway 214 (Qinghai–Yunnan Highway, hereafter G214) from Northern Ela Mountain to Qingshuihe Town, National Highway 109 (Qinghai–Xizang Highway, hereafter G109) from Xidatan to Nagqu, and National Highway 219 (Xinjiang–Xizang Highway, hereafter G219) from Kudi to Shiquanhe Town; the overall transect lengths of G214, G109 and G219 were approximately 400, 750 and 900 km, respectively. Three transects across the entire permafrost regions from north to south in the Eastern, Central and Western TP were established. Many permafrost geological conditions were obtained in the process of the construction and renovation of three highways, and many permafrost roadbed monitoring sections were subsequently set along the highways (Jin et al., 2008; Sheng et al., 2015). Based on these background data and our investigated results (geophysical and drilling exploration), the permafrost distribution limits and geothermal features of three transects were generated and used as the validation datasets.

## 2.2 Spatial datasets

### 2.2.1 Existing two benchmark permafrost maps

The mostly widely used permafrost distribution benchmark maps are 1) The *Map of Permafrost on the Qinghai–Tibetan Plateau*, which was compiled by Lanzhou Institute of Glaciology and Geocryology, Chinese Academy of Sciences (hereafter TP-1996) to support basic research on cryospheric dynamics in China (Li and Cheng, 1996). TP-1996 synthesizes field data, literature, aerial photographs, satellite images and many relevant maps and shows that the area of permafrost is $1.41 \times 10^6 \, \text{km}^2$. 2) The *Map of the Glaciers, Frozen Ground and Deserts in China* was compiled by Cold and Arid Regions Environmental and Engineering Research Institute, Chinese Academy of Sciences (hereafter TP-2006) (Wang et al, 2006). In this map, the permafrost distribution was generated using a 0.5 °C MAGT isotherm as a threshold, which shows that the area of permafrost is $1.12 \times 10^6 \, \text{km}^2$. The MAGT was interpolated based on the relationship between





elevation/latitude and the MAGT observation from all 76 boreholes along the Qinghai–Xizang Highway (Nan et al,
        2002).

## 2.2.2 MODIS LST products

The MODIS LST data used in this study were the 1 km gridded clear-sky MOD11A2 (Terra MODIS) and MYD11A2
(Aqua MODIS) products (reprocessing version 5), which span from 2003 to 2012. Both MOD11A2 and MYD11A2
provide two observations (daytime and nighttime), which means that there are four LST observations for the same pixel
per day. The temporal resolution of MOD11A2/MYD11A2 was 8 days, the LST values represent the 8-day average LST
values (the missing values were ignored in the calculation) (Wan, 2009; Wan and Dozier, 1996), and there are
theoretically 46 groups of LST values every year. While the 8-day MODIS LST products have more reliable data than
daily products, they still have massive missing values when establishing the mean daily LST empirical models due to
clouds or other factors (Prince et al., 1998). In this study, the Harmonic ANalysis Time-Series algorithm (HANTS)
algorithm was applied to MODIS LST on a per-pixel basis for the entire study area. This was accomplished using the
HANTS    software    developed    by    the    National    Aerospace    Laboratory    (NLR)    of    the    Netherlands
(http://gdsc.nlr.nl/gdsc/en/tools/hants). The five parameters needed for the HANTS analysis (Roerink et al., 2000) were
set as follows: the high/low suppression flag (SF), "low" because the presence of undetected clouds will lower the
temperature of the entire pixel, thereby reducing the surface temperature; the range of valid values, 243–323 K; the fit
error tolerance (FET), 6 K; the degree of over determinedness (DOD), 7; and the number of frequencies (NOF), 2 (Xu
et al., 2013).

The MODIS LSTs represent instantaneous observation values, and the overpass times of the satellites do not accurately
correspond to standard meteorological observation times (Beijing time: 2:00, 8:00, 14:00, and 20:00) (China
Meteorological Administration, 2003). Therefore, an arithmetic average of the four LST observations with the same
weights will produce a large deviation from the mean daily LST (Wang et al., 2011). In this study, a multiple linear
regression was employed to distribute different weights to each MODIS LST observation to establish the mean daily
LST empirical model. The details of processing are described in the reference Zou et al. (2014). In this study, the
empirical formula is as follows:

$$LST_{daily} = 0.18 \times Terra_{day} + 0.269 \times Terra_{night} + 0.143 \times Aqua_{day} + 0.435 \times Aqua_{night} + 0.896 \qquad (1)$$

where $LST_{daily}$ is the mean daily LST, $Terra_{day}$ is daytime LST observation of MOD11A2, $Terra_{night}$ is nighttime LST
observation of MOD11A2, $Aqua_{day}$ is daytime LST observation of MYD11A2, and $Aqua_{night}$ is nighttime LST
observation of MYD11A2.

The calculations of the thawing indices (Degree Days of Thawing, DDT) and freezing indices (Degree Days of Freezing,
DDF) were based on the 8-day average LST calculated from the previous processing. The procedures were realized using





the IDL programming language. To maintain the stability of the data, the DDF and DDT from 2003 to 2012 were obtained and averaged as the model inputs.

### 2.2.3 Soil thermal properties

The spatialization of soil thermal parameters was realized according to the results of the soil types. The classification of
soil types was performed using the Decision Tree See 5.0 software and the Soil-Land Inference Model (SoLIM) in conjunction with soil type and environment factor data (Li et al., 2014; Li et al., 2015b). According to the Soil Taxonomy System, there are five soil orders on the TP as follows: Gelisols, Aridisols, Mollisols, Inceptisols, and Entisols. Considering the availability of soil sample parameters, the characteristics of sampling regions, and model applicability, the empirical model of soil thermal conductivity proposed by Kersten (1949) was adopted in this study. The equation of
thawed soil thermal conductivity is

$$k_t = 0.1442 \times (0.7 \times log\omega + 0.4) \times 10^{(0.6243 \times \gamma_d)} \tag{2}$$

Furthermore, the equation of frozen soil thermal conductivity is

$$k_f = 0.01096 \times 10^{(0.8116 \times \gamma_d)} + 0.00461 \times 10^{(0.9115 \times \gamma_d)} \times \omega \tag{3}$$

where $k_t/k_f$ is the thermal conductivity (W m$^{-1}$ K$^{-1}$) of thawed/frozen soil, $\omega$ is the soil moisture content (%), and $\gamma_d$ is
the soil bulk density (kg m$^{-3}$). Both $\omega$ and $\gamma_d$ were measured via soil samples collected in the field survey. The soil samples were classified according to soil orders; moisture content and bulk density values were averaged within soil orders to eliminate abnormal values (Table 2, the values show the mean with standard deviation of soil thermal parameters of each type).

### 2.2.4 Glacier and lake data

The spatial distribution and area of glacier and lake data on the TP were from the Second Glacier Inventory Dataset of China (Guo et al., 2014) and the Cryosphere Information System (Li, 1998) provided by Cold and Arid Regions Science Data Center (http://westdc.westgis.ac.cn).

### 2.3 TTOP model

Considering the model's usefulness and sophistication, spatial scales and available datasets (Riseborough et al., 2008),
we selected the Temperature at the Top Of Permafrost (TTOP) model (Smith and Riseborough, 1996) to simulate the permafrost distribution on the TP.

The TTOP model can be expressed as follows:



$$TTOP = \frac{k_t/k_f \times DDT - DDF}{P} = \frac{(r_k \times n_t \times I_t) - (n_f \times I_f)}{P} \qquad (4)$$

where $P$ is the annual period (365 days), $DDT$ ($n_t \times I_t$) is the ground surface thawing indices, and $DDF$ ($n_f \times I_f$) is the ground surface freezing indices. $n_t$ and $n_f$ are n factors of the thawing and freezing seasons, and $I_t$ and $I_f$ are the air temperature thawing and freezing indices, respectively. $r_k = k_t/k_f$ is defined as the ratio of the thermal conductivity coefficient when soil is thawing and freezing, and it is tightly related to soil properties.

From Equation 4, if the $DDF$ is greater than $DDT \times k_t/k_f$ ($n_f \times I_f > r_k \times n_t \times I_t$), $TTOP$ will be below 0 °C, and permafrost exists. This processing was realized in the ArcGIS software program with the following expression:

$$D = \begin{cases} 1, & TTOP \leq 0 \quad permafrost \\ 0, & TTOP > 0 \quad seasonally\ frozen\ ground \end{cases} \qquad (5)$$

In this study, the land surface temperature was directly used as the upper boundary conditions in the model; therefore, the LST calculation procedure with air temperature and n factor was omitted. To compare with TP-1996 and TP-2006 conveniently, the result of permafrost distribution simulated by the TTOP model was abbreviated to TP-2016. The regions of glacier and lake were excluded from the permafrost distribution modelling of the TTOP model. In addition to permafrost and seasonally frozen ground, unfrozen ground was also identified in this study. The unfrozen ground was defined as the region where the extreme minimum LST ≥ 0 °C. The night Aqua MODIS LST (observation time approximately 3:00 a.m.) was employed as input data for the determination of unfrozen ground area.

## 2.4 Accuracy evaluation

The permafrost distribution of the borehole locations, five IRs and three transects were employed to estimate the accuracies of the three maps (TP-1996, TP-2006 and TP-2016). First, the spatial distribution of borehole temperature data across a permafrost domain or seasonally frozen ground area has been used as the criterion of advantages and disadvantages of results for three time snapshots of 1996, 2006 and 2016. The permafrost distribution across the five IRs and three transects were selected as the real values to validate the three maps.

To evaluate the agreement of the simulated permafrost distribution and the observed results, the kappa coefficient (K) (Cohen, 1960), which measures the degree of agreement, was selected for accuracy evaluation.

$$K = \frac{s/n - (a_1b_1 + a_0b_0)/n^2}{1 - (a_1b_1 + a_0b_0)/n^2} \qquad (6)$$

where the total number of pixels is $n$, and $s$ is the number of pixels in which the simulation and investigated results agree. The number of investigated result pixels with permafrost is $a_1$, and those without are $a_0$, and the simulated map pixel



numbers are $b_1$ and $b_0$. Empirically and statistically arbitrary quality values for K have been proposed; e.g., Cohen (1960)
suggested that K ≥ 0.8 signifies excellent agreement, 0.6 ≤ K < 0.8 represents substantial agreement, 0.4 ≤ K < 0.6
represents moderate agreement, 0.2 ≤ K < 0.4 represents fair agreement, and a lack of agreement corresponds to K < 0.2.

## 3 Results

### 3.1 Permafrost distribution modelling of TTOP

Fig.2 shows the simulated permafrost distribution of the TTOP model on TP (TP-2016). The distribution areas of
permafrost and seasonally frozen ground were $1.06 \times 10^6$ km$^2$ and $1.46 \times 10^6$ km$^2$, excluding glaciers and lakes, which
account for 40% and 56% of the total TP area, respectively. The result shows that the permafrost distribution was centred
in Southern Qinghai and Northern Tibet. The Northern Qiangtang Plateau and Kunlun Mountain were the regions with
the most permafrost developed regions which extends west and northwest to Karakoram Mountain. As the elevation
decreases and the ground temperature increases with increasing distance from the central region, the permafrost
continuity decreases gradient. The geographic north and south boundary of permafrost were Xidatan and Anduo from
the mark sites of Qinghai–Xizang Highway. There were a few areas of permafrost in the high mountains from Anduo to
the Southern Tibet Valley. Due to the existence of the Bayan Har Mountains and Anemaqen Mountain, whose elevations
are above 5000 m, there are permafrost occurrence in the Eastern TP. Some unfrozen ground exists in the southeast
margin of the TP, whose area is approximately $0.03 \times 10^6$ km$^2$, which account for 1% of the total TP area.

Due to the negative temperature and high soil moisture at the bottoms of glaciers, permafrost generally developed
beneath glaciers (Harris and Murton, 2005). On the TP, there is approximately $0.04 \times 10^6$ km$^2$ of glaciers according to the
second glacier inventory dataset of China (Guo et al., 2014); therefore, the permafrost distribution area should be
$1.10 \times 10^6$ km$^2$, including glaciers. TTOP is formulated with respect to the ground surface temperature. However, the
satellite-derived LST observes a mixture of the vegetation canopy, snow cover, and ground surface, which depend on
the region and the season. The snow cover distribution is spatially quite variable over the TP due to the complex terrain;
the most persistently snow-covered areas occur in the southern and western edges of the TP. In the interior of the TP, the
snow cover fractions are relatively small and less persistent (Pu et al., 2007), which was beneficial for applying remote
sensing LST. In the Western TP, the permafrost regions mainly underlie alpine desert and bare area, and the MODIS LST
basically reflects the ground surface temperature. In the Eastern TP, the alpine meadow is the dominant vegetation type,
and the temperature derived from MODIS sensors is essentially the temperature mixture of the vegetation canopy and
ground surface. Considering that the height of vegetation is relatively low in permafrost regions and that the thermal
infrared band has some penetrability, we assumed that the MODIS LST can represent the ground surface temperature in
the growing season in this study.

### 3.2 Validation with borehole observations

Boreholes are the most convincing evidence to determine whether permafrost exists. Fig.3 shows the spatial distribution



of borehole locations at permafrost or seasonally frozen ground in five IRs of three maps. Different combinations were set up to analyse the difference of the three results; columns a, b, and c show the results of TP-1996, TP-2006, and TP-2016, and rows 1, 2, 3, 4, and 5 show the results of XKL, GZ, AEJ, B-Q, and WQ IRs, respectively. The result shows that TP-1996 is insensitive to the geographical boundaries across all five IRs, and there are plenty of erroneous
interpretations of both permafrost and seasonally frozen ground. TP-2006 has higher sensitivity to the boundaries, especially in WQ IR; however the recognition of the other four IRs is not good enough, and the areas of permafrost distribution are overestimated. Compared to TP-1996 and TP-2006, TP-2016 performed better at identifying the geographic boundary of permafrost distribution, identifying almost all the boundaries of the five IRs correctly, especially for the thawing regions in the valley of the Northwestern XKL IR (Fig.3 c1) and that around the lakes of the Eastern AEJ
IR (Fig.3 c3). TP-2016 featured some misjudgment, mainly affected by local terrain factors including the seasonally frozen ground distributed in valleys and a few permafrost spots at the margin, such as the two seasonally frozen ground boreholes in the Northern AEJ IR (Fig.3 c3) and three permafrost boreholes at the southwestern limit of GZ IR (Fig.3 c2).

### 3.3 Validation with five investigated regions (IRs)

The permafrost distributions of five IRs were employed as truth values to validate the modelling results of three maps to analyse their performance in terms of geographical boundary recognition ability. TP-1996 performed worst at recognizing the boundaries of permafrost in five IRs; it misidentified all boundaries, with a low kappa coefficient (K < 0.2), due to more misjudgment or overestimated permafrost pixels. TP-2006 also performed poorly in the XKL, GZ, and AEJ IRs (K < 0.2) but performed better in the B-Q and WQ IRs, with a kappa coefficient reaching 0.63 and 0.77. TP-
2016 had poor performance in the AEJ IR; the kappa coefficient reached only 0.38, which is an improvement to some extent over that of the former two. In addition, it represents moderate agreement with the XKL and GZ IRs and substantial agreement with the B-Q and WQ IRs, whose kappa coefficients were 0.54, 0.48, 0.68 and 0.78, respectively. The average accuracies of TP-1996, TP-2006 and TP-2016 were 0.06, 0.35 and 0.57, respectively. TP-2016 performed best in the validation with the investigated permafrost distribution from both the individual and averaged accuracies of five IRs
(Table 3).

The results of the AEJ IR and surrounding area are selected to compare the differences among the three maps (Fig.4). In the AEJ IR, the investigated result shows that the seasonally frozen ground is mainly distributed at the northern valley and the Eastern Ayakekumu Lake surrounding areas and features permafrost. TP-2006 shows all judgement for permafrost in the AEJ IR, which obviously overestimated the area of permafrost. Although TP-1996 shows some
seasonally frozen ground in the Northwestern AEJ IR, the locations were misjudged. TP-2016 judged approximately 30% seasonally frozen ground in the Northern and Eastern AEJ IR. Although the correct pixels were few, the locations were correct, especially in the eastern part, just at the geographic boundary of permafrost. The observed MAGT of the borehole closest to Ayakekumu Lake was 3 °C, which means that seasonally frozen ground exists there, and TP-2016 accurately modelled this phenomenon. In the regions around the AEJ IR, TP-2016 simulated the thawing regions around Aqikekule



Lake (area approximately 350 km$^2$) and its supply river, and this did not appear in the other two maps. Most lakes on TP are formed due to tectogenesis; the major axis basically remains consistent with the main structure lines and the secondary level fracture direction, and there generally exists penetrative or nonpenerative thawing regions under and around tectonic lakes (Zhou et al., 2000). TP-2016 also shows seasonally frozen ground in the mountainous region proximate to the Pitileke River, while the other two maps did not identify that. TP-2016 was more accurate in this respect; the ground temperature was affected by the higher temperatures of the water bodies, resulting in the appearance of seasonally frozen ground.

The permafrost distribution of TP-1996 and TP-2006 was modelled according to the relationship between temperature and three-dimensional zonalities (longitude, latitude, and elevation) (Cheng, 1984). The higher weight of elevation from the regression equation determined that it has greater influence than that of longitude and latitude when interpolating temperature (air temperature or MAGT). The high continuity and low variability of the elevation difference in permafrost regions lead the results to appear more continuous; however, the temperature differences caused by local factors (e.g., lakes or rivers) are masked to a large degree and thus result in an excessive occurrence of the lower extrapolated temperature; this could be used to explain the overestimated area of permafrost distribution in the previous TP-1996 and TP-2006. The application of the remote sensing data, which overcomes this shortcoming, can better reveal the spatial heterogeneity of LST. Relative to the two benchmark maps, the result of TP-2016 driven by the processed MODIS LST in this paper is very sensitive to thawing regions formed by surface water, and the results show that there are many thawing regions surrounding lakes and major rivers that corresponding to the previous studies (Lin et al., 2011; Niu et al., 2011).

### 3.4 Validation with three transects

The permafrost distribution of three transects (G214, G109 and G219) of three maps were extracted to compare to the investigated results to comprehensively evaluate their performance on the mainly permafrost developed regions on the TP. The accuracy statistics of three maps in the three transects are listed in Table 4. TP-1996 has the worst accuracy in the three maps with an average kappa coefficient of 0.34. The accuracy of TP-2006 is higher than that of TP-1996 with an average kappa coefficient of 0.50; it performed well especially in transect G109. TP-2016 has the highest accuracy; the kappa coefficients are 0.62, 0.69, and 0.74 for G214, G109, and G219, respectively, with an average of 0.68. TP-2016 performed best in the validation with the investigated permafrost distribution from both the individual and averaged accuracies of the three transects. In the three transects across all permafrost regions from north to south in the Eastern, Central and Western TP, which include most permafrost distribution characteristics in TP, the validation results should be a synthetic evaluation of the three maps.

Fig.5 shows the distributions of permafrost and seasonally frozen ground along the G109 transect of the three maps and investigated result; the elevation and mark sites were also added for analysis. To conveniently compare, the G109 transect was divided into five segments according to the investigated result as follows: two continuous permafrost regions (from XDT to Southern FHSYK, and Southern YSP to Northern AD), one region of seasonally frozen ground only (from



Southern LDH to NQ) and two regions in which permafrost and seasonally frozen ground coexist (from WL to YSP, and AD to LDH). The comparison shows that the three maps performed well in two continuous permafrost regions; almost all permafrost is identified correctly except for several seasonally frozen ground areas in CMEH and BLH of TP-1996. In the region of seasonally frozen ground only, TP-1996 judged permafrost from AD to NQ, which is different from the investigated result and overestimated the permafrost area in this region. TP-2006 and TP-2016 identified that only seasonally frozen ground exists in this region, which is highly consistent with the investigated result. In two regions where permafrost and seasonally frozen ground coexist, a large difference occurred between the three maps and the investigated result. TP-2006 shows that continuous permafrost exists from XDT to Northern AD, performing poorly in the recognition of the thawing regions, and thus overestimating the area of permafrost in the G109 transect. TP-1996 performed better than TP-2006, which recognized some of the thawing regions in TTH, TTH', YSP and AD; however, the widths and locations reveal bias from that of the investigated result. TP-2016 identified almost all thawing region locations correctly with a lower distance difference, which is more consistent with the investigated result than the former two. It is worth mentioning that TP-2006 and TP-2016 identified sporadic permafrost in LDH, which was generally expected as the southern limit of permafrost in previous studies.

In the G109 transect, thawing regions mainly exist due to the surface water effects, regional geologic structure/geothermal effects and penetration/radiation effects, which cause a discontinuity in the plane and depth of continuous distribution of permafrost (Zhou et al., 2000). Due to the large streamflow and high water temperature of TTH, TTH' and Buqu (flow through YSP) rivers, the penetrative thawing regions not only developed on the riverbed and high floodplain, but also expanded to the first or second terrace (width generally reached 5–10 km). Additionally, bare land, gravel layer exists, and a higher mean annual air temperature was beneficial to precipitation infiltration, which created active thermal transfer conditions; therefore, the thawing regions in TTH and YSP were also affected by penetration/radiation effects. However, for the rivers with less streamflow and higher latitude, such as the CMEH and BHL rivers, the nonpenetrative thawing regions are much smaller (generally < 100 m) and thus almost impossible to identify. The thawing regions in Northern WL were mainly affected by regional geologic structure/geothermal effects, which has been validated by the results of engineering geologic surveys of the Qinghai–Xizang Highway and Railway (Jin et al., 2008). From the kappa coefficients of the three maps and investigated result (Table 4) along the G109 transect, TP-2016 can better identify the thawing regions of several kilometres in width caused by local factors (surface water, geothermal, and permeate/radiation effects).

### 3.5 Spatial difference among the three maps

The kappa coefficients of each pair among the three maps were calculated (Table 5) to analyse the spatial difference. TP-1996 revealed low consistency with both TP-2006 and TP-2016; the kappa coefficients were 0.56 and 0.53, respectively, which indicates a large difference. TP-2006 has a substantial agreement with TP-2016; the kappa coefficient reached 0.71. The spatial difference between each pair among the three maps were compared (Fig.6). Compared with TP-2006 and TP-2016, TP-1996 overestimated the permafrost area, which was mainly distributed at the Southeastern TP, south



margin of continuous permafrost, and predominantly continuous and island permafrost in the Southern TP. In addition, TP-1996 misjudged some seasonally frozen ground on the continuous permafrost edge and the thawing regions in the interior TP. The permafrost distribution area of TP-2006 was close to that of TP-2016; the difference mainly exists at the thawing regions in the interior TP, south margin of continuous permafrost, and surrounding regions of the Bayan Har Mountains and Eastern Nyainqntanglha Mountains.

## 4 Discussion

The dataset used in the earliest maps, represented by TP-1996 which has been widely recognized as the benchmark map of permafrost distribution on the TP (Cheng and Wu, 2007; Yang et al., 2010; Mu et al., 2015), were the air temperature, field data, aerial photographs, satellite images and many relevant maps (Tong and Li, 1983; Shi and Mi, 1988; Li and Cheng, 1996). The boundary criteria was mainly based on the air temperature isotherm, and a few modulations were conducted in some regions with permafrost investigations, such as Qinghai–Xizang Highway, Qinghai–Yunnan Highway and Hengduan mountains by the authors with their knowledges (Li and Cheng, 1996). The threshold of the air temperature isotherm was determined by the empirical statistical relationship between the permafrost boundary and the meteorological observations in the Eastern TP (Li and Cheng, 1996); therefore, the universality of the threshold in the Western TP is questionable where data were insufficient and permafrost boundary was determined according to the author's judgement. In addition, high uncertainty exists in the air temperature interpolation because monitoring sites are scarce, are unevenly distributed (more in the Eastern TP and less in the Western TP; more in lower elevation and less in higher elevation) and exhibit poor representativeness; thus, the accuracy of extrapolated air temperature has generally been low (Li et al., 2003; Lin et al., 2002). The permafrost maps were compiled with conventional cartographic techniques that plot the permafrost boundaries on the topographic maps by hand (Tong and Li, 1983; Shi and Mi, 1988; Li and Cheng, 1996), the artificial error depends on the mapper's knowledge and skill is very difficult to control. All of these factors lead to high uncertainty in this map; although it describes "the most accurate" permafrost distribution at that time. Later studies that used this map as a benchmark to compare with the modelling result seems inappropriate today. Actually, these maps are more emphasized on the concept of permafrost regions, which overestimated the permafrost areas too much (Wang et al., 2016). Recently, a similar study to establish the relationship between the mean annual air temperature and the permafrost occurrence probability shows that there is area of about $1.0 \times 10^6$ km$^2$ permafrost on TP which close to that of this study although it cannot indicates detailed permafrost distribution pattern (Gruber, 2012). TP-2006 mapped the permafrost distribution based on the MAGT through considering the characteristics of high altitude permafrost. The spatial ground temperature was interpolated based on the relationship between elevation/latitude and the MAGT from boreholes along the Qinghai–Xizang Highway (Nan et al, 2002) which has high representative in the Central however low in the Eastern and Western TP, that has been demonstrated in the validation of the three transects. In the mapping of TP-2016, the remote sensing data and observations based on an investigation were used to generate the map. In view of the high spatiotemporal resolution and sensitivity to spatial temperature heterogeneity of MODIS LST data, it can reflects accurately the spatial pattern of LST on the TP. In addition, the MODIS



LST data was calibrated by the ground-based LST observations obtained by automatic weather stations in typical permafrost regions which has high representativeness of permafrost climate conditions (Zou et al., 2014), the calibrated temperature is more corresponding to actual condition than the interpolated air temperature. The subsurface thermal properties, which derived from soil investigation data, were also considered in the TTOP model which didn't appear in other studies. The improvement of upper boundary conditions of permafrost model and the employed of massive reliable in situ observed datasets make the high modelling accuracy achieved.

In the earliest maps, there is only some observed data with the field sites along Qinghai–Xizang Highway (Tong and Li, 1983; Shi and Mi, 1988; Li and Cheng, 1996), but little validation in the other regions which means large difficulty in the map evaluation. The threshold of 0.5 °C MAGT isotherm of TP-2006 was determined by the sensitivity analysis to compare with the permafrost distribution pattern of TP-1996 without any other independent validation data (Nan et al, 2002). The validation in this study shows that the accuracy of TP-2006 was higher than that of TP-1996; however, it highlights the excessive elevation effects when interpolating the ground temperature, thereby masking the effects of local factors to some degree. The better performance in the B-Q and WQ IRs may be because these two IRs were close to the Qinghai–Xizang Highway and have similar geomorphology to the highway relative to the other three farther IRs (XKL, GZ and AEJ IRs). This suggests that the MAGT model would reflect the actual distribution of permafrost with enough typical borehole ground temperature observations, and that is why we use it to modelling the permafrost distribution of five IRs. From the spatial representativeness, the validation results of five IRs emphasized the performance on recognizing boundary and that of three transects specially emphasized the overall evaluation of three maps. The validation results of both five IRs and three transects show that TP-2016 performed the best and achieved the highest accuracy in the three maps. The result could provide a standard permafrost map on the TP in contemporary climate. The data processing method and procedure provides a new concept on permafrost distribution modelling, which could be used for other places.

Although TP-2016 performed better than TP-1996 and TP-2006 and showed substantial agreement with the investigated results, it still results in some misjudgments. The investigation results of five IRs considered more detailed local factors that influenced the permafrost distribution with higher spatial resolution whose scales do not match those of the three maps; this would be a major error source when validated with IRs. In addition, the extent size of the IRs determines the accuracy of the validation to suitable extents exhibiting higher accuracies. To ensure the correct permafrost distribution of IRs, the extents in this study were determined arbitrarily via the locations of boreholes with the minimum extents, which means that the greatest difference was exhibited when comparing; this would lead to lower similarity and kappa coefficient. Hence, a more accurate determination of extents of IRs may benefit from more rational validation accuracies, which also demonstrate that the accuracies in this study are reasonable; this suggests that more detailed conditions should be considered in future studies.





## 5 Conclusions

This study exploits the advantages of the high spatiotemporal resolution of MODIS LST products to construct a database of mean daily LST of the TP. The permafrost distribution is simulated by the TTOP model combined with ground observation and soil investigated datasets, and the model was validated against the permafrost distribution obtained from the borehole temperature data, five IRs and three transects and compared to two recent benchmark maps. In compliance

with borehole temperature data, the suggested method of permafrost boundary identification shows a better result than the two maps, especially for the thawing regions in valleys and around lakes. The accuracy of method validation shows that the TP-2016 case has the highest kappa coefficients for both five IRs and three transects. The average coefficients are 0.57 and 0.68, respectively. The modelling estimation shows that $1.06 \times 10^6$ km$^2$ of permafrost, $1.46 \times 10^6$ km$^2$ of seasonally frozen ground, and $0.03 \times 10^6$ km$^2$ of unfrozen ground could be on the TP. Compared with two recent

benchmark maps, the TTOP model is superior in recognizing thawing regions, especially in the areas surrounding lakes and rivers. The new permafrost distribution map represents a promising basic dataset for further permafrost research. With the advantages of dynamic real-time monitoring of LST changes, MODIS LST products can also be utilized for the study of permafrost ground temperature and active layer changes.

**Acknowledgements.** We thank the Level 1 and Atmosphere Archive and Distribution System for providing MODIS

land products, and Cold and Arid Regions Science Data Center for data of glacier and lake on the Tibetan Plateau. This research was financial supported by the National Major Scientific Project of China "Cryospheric Change and Impacts Research" (2013CBA01803), the Creative Research Groups of National Natural Science Foundation of China (No. 41421061), the key project of the Chinese Academy of Sciences (KJZD-EW-G03-02). We would also gratefully acknowledge Professor Jerry Brown, Stephan Gruber and Sergey Marchenko for their helpful and constructive

suggestion on the manuscript.



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



**Table 1.** Field survey samples statistics in five investigated regions

| | Investigated Region (IR) | | | | | |
| | WQ | B-Q | AEJ | GZ | XKL | Total |
|---|---|---|---|---|---|---|
| Boreholes | 21 | 40 | 13 | 23 | 28 | 125 |
| Soil pits | 74 | 55 | / | 19 | 51 | 199 |


**Table 2.** Soil thermal parameters of each type on the Tibetan Plateau

| Soil order | Samples Number | Moisture content (%) | Bulk density (kg m$^{-3}$) | thawed soil thermal conductivity (W m$^{-1}$ K$^{-1}$) | frozen soil thermal conductivity (W m$^{-1}$ K$^{-1}$) |
|---|---|---|---|---|---|
| Aridisols | 43 | 7.76 (±3.0) | 1601.9 (±173.2) | 1.47 (±0.42) | 1.25 (±0.63) |
| Entisols | 10 | 8.79 (±6.64) | 1447.7 (±164.8) | 1.23 (±0.17) | 1.01 (±0.33) |
| Gelisols | 56 | 22.24 (±13.79) | 1277.6 (±310.0) | 1.22 (±0.36) | 1.62 (±0.44) |
| Inceptisols | 94 | 16.22 (±7.37) | 1313.4 (±221.7) | 1.18 (±0.34) | 1.30 (±0.53) |
| Mollisols | 14 | 20.00 (±5.66) | 1186.9 (±141.3) | 1.05 (±0.23) | 1.22 (±0.48) |



**Table 3.** Kappa coefficient statistics in five investigated regions of three maps

| Investigated Region | TP-1996 | TP-2006 | TP-2016 |
|:---:|:---:|:---:|:---:|
| WQ | 0 | 0.77 | 0.78 |
| B-Q | 0 | 0.63 | 0.68 |
| AEJ | 0 | 0 | 0.38 |
| GZ | 0.15 | 0.19 | 0.48 |
| XKL | 0.14 | 0.17 | 0.54 |
| Average | 0.06 | 0.35 | 0.57 |


**Table 4.** Kappa coefficient statistics in three transects of three maps

| Transect | TP-1996 | TP-2006 | TP-2016 |
|:---:|:---:|:---:|:---:|
| G214 | 0.32 | 0.41 | 0.62 |
| G109 | 0.21 | 0.59 | 0.69 |
| G219 | 0.47 | 0.49 | 0.74 |
| Average | 0.34 | 0.50 | 0.68 |


**Table 5.** Kappa coefficient statistics among three maps

| | TP-1996 | TP-2006 | TP-2016 |
|:---:|:---:|:---:|:---:|
| TP-1996 | 1 | 0.56 | 0.53 |
| TP-2006 | / | 1 | 0.71 |
| TP-2016 | / | / | 1 |






**Figure 1.** Spatial distribution of the field survey regions on the Tibetan Plateau





**Figure 2.** Spatial distribution of frozen ground on the Tibetan Plateau







**Figure 3.** Spatial distribution of boreholes in five IRs of three maps





**Figure 4.** Comparison of the three maps in and around the AErJin investigated region (a: TP-1996; b: TP-2006;
c: TP-2016)





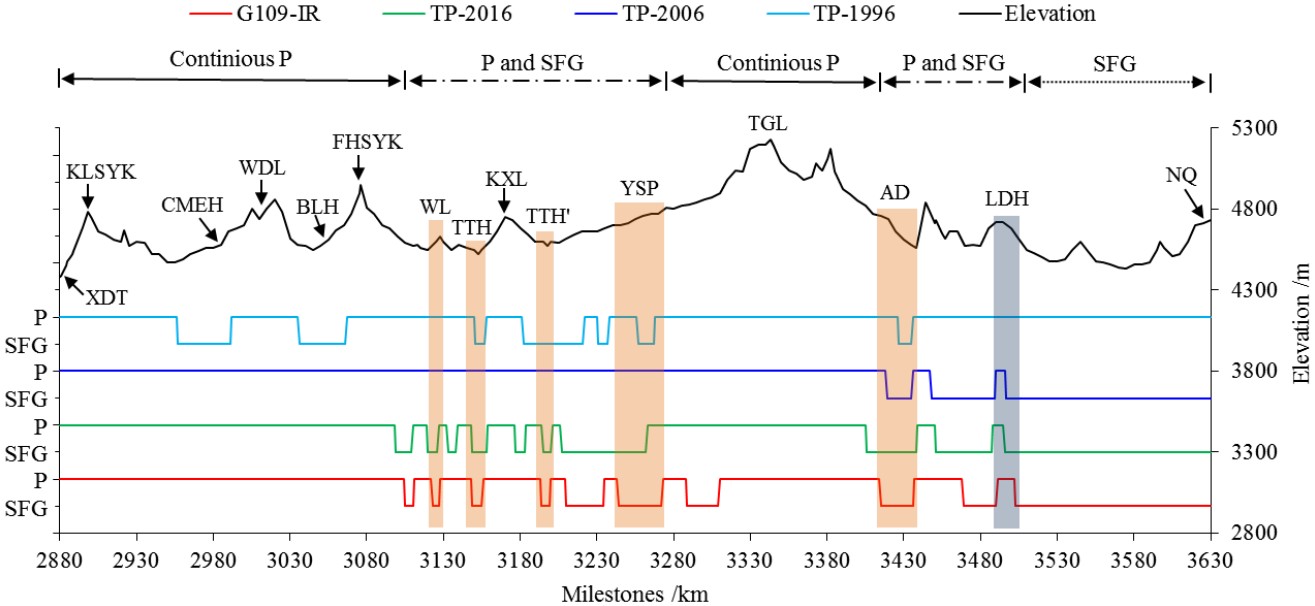

**Figure 5.** Comparison of permafrost distribution of three maps along the G109 transect with investigated result (P: permafrost, SFG: seasonally frozen ground; XDT: Xidatan, KLSYK: Kunlun Mountain Peak, WDL: Wudaoliang, BLH: Beilu River, FHSYK: Fenghuo Mountain Peak, WL: Wuli, TTH: Tuotuo River, KXL: Kaixin Mountain Ridge, TTH': Tongtian River, YSP: Yanshiping Town, TGL: Tangula Mountain Peak, AD: Anduo Town, LDH: Liangdaohe, NQ: Nagqu Town, G109-IR: investigated result of permafrost distribution in G109 transect)





680

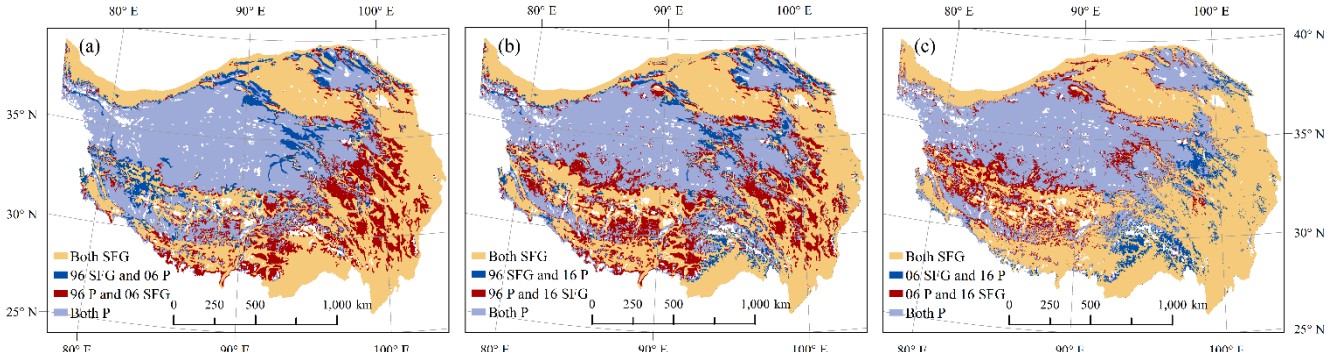

**Figure 6.** Spatial difference among the three maps (96: TP-1996, 06: TP-2006, 16: TP-2016; SFG: seasonally frozen ground, P: permafrost)