# Peer review of "A New Map of the Permafrost Distribution on the Tibetan Plateau"

_The Cryosphere, 2016_

## Referee Comment (RC1) · Anonymous Referee #1 · 4 Jan 2017

Apologies to all for late post.

**Summary**

This manuscript constructs and evaluates a new permafrost map for the Tibetan Plateau (TP). The map is created by using MODIS LST and soil types based on in-situ observations and an empirical model. These data are used to drive the TTOP model (Brown & Riseborough 1996) to determine spatial distribution of permafrost (defined as TTOP lessthan 0degC). The map is evaluated against extensive datasets of ground temperature measurements from boreholes, infrastructure monitoring systems and previous field campaigns. The new map is also compared to two previous generations of regional permafrost maps. This manuscript presents an interesting attempt to combine remote sensing, observational and modelled data to improve understanding of permafrost distribution on the Tibetan Plateau - I think this is a good approach and useful contribution. However, I think several issues need to be dealt with more rigorously prior to publication.

**Main comments**

1. LST obviously is not equal to ground surface temperature (GST) that is required to drive TTOP. You acknowledge this in Section 3.1, however: (A) this needs to be much more explicit as you describe your methods. (B) Does it not make it extremely difficult to interpret values obtained under snowcover as snow surface temperatures are likely to be much lower than GST even in shallow (albeit likely cold and dry, therefore lower thermal conductivities than a temperate snowpack) snowpacks of the TP, especially on clear nights where high emissivities will cool the snow surface much more than the GST. I think its really important you at the very least quantify how significant this problem will be in your study region and probably try to introduce a term that accounts for the offset between GST and surface temperatures under snow on the ground conditions. Additionally I think this statement is wrong:

"In this study, the land surface temperature was directly used as the upper boundary conditions in the model; therefore, the LST calculation procedure with air temperature and n factor was omitted."

LST is likely to be very similar to near surface air temperature - this approach therefore ignores the n-factors which are important in describing the offset between air and ground surface conditions particularly under snowcover as described above, also effects of vegetation (less significant perhaps).

2. You don't give any evaluation of the MODIS product - how well does it perform in the region? WHat are the uncertainties under snowcover (wet/dry), vegetation, arid soils etc. You have AWS data from the permafrost field campaign you mention in Section 2.1 which may give you some clue if you measure surface temps (of coarse point/spatial scaling needs to be acknowledged). Cite the literature that has looked at uncertainties

in MODIS LST regionally/globally and give some incites on how you expect this to affect your model setup. This forms the basis of your method and is therefore really important to critically discuss.

3. MODIS LST is a coarse resolution 1 km product. This of course will make any kind of discrimination of permafrost units at the subgrid scale difficult - mainly important on the north slope of Himalaya and other mountain regions of complex topography on the TP. It should be thoroughly discussed what limitations this poses for your results.

4. Bedrock and debris slopes do not seem to be included in your "soil" classes and presumably are important land classes in your region. How do you deal with these?

5. Why not present actual values derived from the TTOP model instead of just a binary map? This would be interesting to see eg. where warm/cold permafrost exists.

6. How do you incorporate the effect of solar radiation (slope, aspect + possibly horizon, sky view factor) into your five investigated regions (IR) that you use as validation? From reading section 2.1.2 it appears that you determine a lower limit of permafrost (LLPs) and extrapolate this across your IR. However this sentence:

"The permafrost map was generated for each IR based on the criteria of LLPs in different conditions combined the digital elevation model (DEM) data..."

Suggests you do something which may account for at least aspect. This needs to be well described as forms the basis of you evaluation. We need to know how well we can trust this and what uncertainties are involved in these validation datasets.

7. In the validation exercise with the 2016 map we see large differences in kappa values: 0.38 – 0.78. Some discussion is given (p.14 l.439–448) which as far as I can tell indicates that smaller IR are more accurate due to density of measurements. However, best (0.78, WQ) and worst performing (0.38, AEJ) are roughly the same size. It is important you discuss these differences in performance with respect to how well you think your model performs in the various regions eg. uncertainties such as complex
topography, LST-GST offsets etc.

8. How long are your borehole records? These need to be better described in your data section. You show in Figure 3 how the new map better represent seasonally frozen ground eg. subset a3-c3. But how do we know whether the model is better or simply that the permafrost has thawed in this region over the last 20 years. The borehole measurements are not contemporary with the old permafrost maps as I understand - but that needs to be described as stated above. Additionally, in validation you compare a map derived from 2003–2012 MODIS data with borehole records of possibly another period. Basic point: it seems that comparability of different maps and validation datasets is problematic and these issues should be discussed well.

9. In section 3.5 you compare maps 1996, 2006 and 2016. What is the main message from this comparison? How do you disentangle changes in permafrost distribution computed from possible actual changes in MAAT over the last 20 years and differences due to different methods and sources of uncertainty? It would be good to be clearer about what the various differences that are observed are correlate with i.e. complex topography, latitude, data scarcity etc.

10. Issue of permafrost conditions out of equilibrium with todays climate i.e. warming permafrost conditions, should be discussed. Surface forcing could indicate no permafrost according to todays conditions - but there exists a long response time of permafrost bodies to modern atmospheric conditions. Therefore any map based on a contemporary forcing likely underestimates permafrost extent and especially, arguably the most interesting/ disruptive warming/thawing permafrost bodies. This fact does not have an easy solution, but certainly should be discussed.

11. How do you identify 'thawing regions' (Section 5 l.460 and mentioned throughout text). This would require some form of transient modelling that demonstrates a transition from permafrost to non-permafrost conditions? As far as I can tell you are equating detection of seasonally frozen ground to thawing conditions. If you use 'sea-

sonally frozen ground' in figures also use this in text, otherwise confusing to reader that likely associates the word 'thaw' with a change in permafrost conditions.

12. Language of Section 4 is very poor in sharp contrast to rest of paper which is generally fairly good.

**Minor comments**

1. Might be worth citing Gruber 2012 (cited later in paper) in your introduction where you discuss TP permafrost maps.

2. p.6 l.174: massive –> numerous.

3. p.6 l.175: describe what HANTS is and why you use it. Details can be left to the reference but reader needs to know the basic purpose of this method.

4. p.6 l.183: what are the MODIS overpass times? How many Swathes used?

5. p.7 eq.4: mention that kt/kf comes from properties derived in Section 2.2.3. Make this link more obvious in text.

6. p.9 l260 "decrease gradient" –> "decreases linearly", is that what you mean?

7. p. 13 l.415: I would rather say medium spatio-temporal resolution. I don't think 4 daily values at 1 km qualifies as 'high res' on either dimension.

8. p.13 l.416: "it can reflects" –> "it can represent".

9. p.14 l421-422: "The improvement of upper boundary conditions of permafrost model and the employed of massive reliable in situ observed datasets make the high modelling accuracy achieved. " –> "The improvement of upper boundary conditions of the permafrost model and the use of large quantities of reliable in situ observed datasets, leads to a high modelling accuracy."

10. Based on comment above about comparability of maps and validation data, I don't think this statement is so straightforward (p14 l439): "Although TP-2016 performed

better than TP-1996 and TP-2006 and showed substantial agreement with the investigated results, it still results in some misjudgments"

11. What is the permafrost distribution of figure 1 based on?

12. Acknowledgments: remove "Level 1 and"

---

## Referee Comment (RC2) · Anonymous Referee #2 · 5 Jan 2017

Review of Zou et al. In this manuscript the authors present a novel approach for characterizing permafrost distribution across the Tibetan Plateau with the commonly used Temperature at the Top of Permafrost model (e.g. TTOP). Comparison with locally collected data and prior maps suggest that the new map provides a better baseline of permafrost distribution in the region than given by previous (more arbitrary maps). The authors use MODIS LST data to force the local climate conditions for the TTOP model.

Overall, I consider the paper to be of sufficient quality to be published in The Cryosphere after revisions have been provided. However, I also believe that there are a number of key points that have to be addressed in order for this paper to be accepted for publication. I have gone back and forth between trying to decide whether this should constitute 'major' or 'minor' revisions.

[Figure]

Notably, I consider the following to be major points: [1] MODIS LST The authors use MODIS LSTs as the key input for their model of permafrost distribution. However, MODIS LSTs measure a combination of different surfaces including the snow surface. If, as the authors postulate, there is only minimal snow cover across the region and correspondingly that MODIS LSTs can be used in the winter then this is all fine. However, the authors have not shown conclusively that snow cover impacts on LST retrievals can be ignored for their region. Addressing this point is a necessity for this manuscript to be considered suitable for publication in The Cryosphere. Likewise, there is certainly some effect of canopy cover in the summer which has been ignored by the authors. It would be useful if the authors examined the ecotype related impacts on the LST and correspondingly how this may affect the distribution of permafrost in the region. I also suggest that the authors produce an additional figure which shows a 1st panel with the estimated regional snow depth across the area (either from reanalysis or other datsets) and a 2nd panel that shows the spatial distribution of vegetation classes (broadly) across the region so that as reviewers we can determine the degree to which this issue may be problematic. Another issue with the MODIS LST that I find concerning is that the authors make the claim that MODIS is preferable to interpolation for temperature (it seems to be in the context of air temperature). A number of studies have found issues with MODIS-derived (or aided) air temperature products with only minimal improvements being observed (if at all) in terms of cross-validation.

Although I do think that MODIS products have utility for permafrost purposes, more work must be done to demonstrate that these products offer improvement over high resolution interpolation of station-based temperature products. It is important that the permafrost community ensures that the usage of LSTs from MODIS for driving permafrost models is assessed at each usage given the spatial heterogeneity of the factors influencing MODIS LSTs.

[2] TTOP modelling output The others provide a simple binary term for the presence or absence of permafrost that is useful in the context of total area numbers but also

means that huge amounts of information are unavailable. A map of TTOP temperatures could be useful in interpreting areas most susceptible to future change and also for the purposes of understanding permafrost thicknesses under a variety of environments. I would highly recommend that the authors at least present one map showing the spatial distribution of TTOP temperatures.

[3] Uncertainties Given the uncertainties that may be present in the LST products and in distributing rk across the landscape, it would seem important that some assessment of uncertainty is provided for the estimates of total permafrost area. It also may be a little optimistic to assume that all glacier area would correspond to permafrost area given the vast range of climates in the region. Such an assumption would require a very detailed assessment to rationalize – I'd prefer it be left out.

[4] Non-equilibrium permafrost The authors should consider the results of Riseborough (2007) when evaluating their TTOP model output and particularly in the context of non-equilibrium permafrost. Is the region warming and if so would this be impacting the distribution of permafrost as measured from this equilibrium model? One of the challenges in using a MODIS derived product is that the relatively short period of coverage makes it more challenging to model in hindcast.

Minor points: L16: Remove "mostly". L27-28: Identifying 'thawing regions' seems unclear to me. L38-39: This sentence could use some grammar editing for clarity. L41-42: Urgent is perhaps a bit strong of a word here, as is 'situation'. L46: "there is great variation" -> "there is considerable variation" L49-50: This sentence should be re-written to be clearer. At present, it makes no sense. L51-52: What is the difference between a topographic map and a base map? L54: "On the" -> "on" L55-56: This statement is not true. GIS techniques were used before 2000... L58: What does "stability of elevation" mean? L74-75: I do not agree with this sentence. Temperature and reanalysis data have a higher temporal resolution than MODIS and can be interpolated more accurately. In my experience, MODIS LST products in the Subarctic and Arctic are not suitable alone for characterizing spatial variations in temperature. L80: I agree.

The authors should provide examples of this validation. L87: Remove "plenty of" L89: Remove "perfect" L95: Remove this sentence L96: Remove "combined" L113: What is "drilling method". The grammar seems a bit off. L136-137: The grammar in this sentence should be revised. L157: "mostly widely" -> "most widely" L174: "massive missing values" -> "many missing values" L175: "Harmonic ANalysis Time" -> "Harmonic Analysis Time" L176-177: Remove sentence or combine with earlier sentence L197: What is "stability of the data"? L197: I prefer FDD and TDD or sFDD and sTDD to DDF and DDT. L199: Amend to: "Soil thermal characteristics were modeled according to parameters measured from soil types encountered in the field". L232: We do not need a sentence to tell us that an abbreviation was used. L259-260: Amend: "increases with increasing" and "decreases. . . decreases". L264: This sentence could be shortened with the use of brackets. L280: Boreholes are not "convincing evidence" of permafrost rather they can determine if permafrost exists or not. This sentence should be revised. L311: ". . .correct. . . correct" – please revise L329: "overcomes this shortcoming" – That is not necessarily proven in the study. L360: "lower distance difference" – Please clarify. L389-L392: This sentence is confusing – please revise. L399: "are unevenly" -> "unevenly" L400: What is poor representativeness? L404: "the most accurate" – Remove this sentence. L409: Poor sentence grammar – Please revise. L416: "reflects" -> "reflect" L418: "high representativeness" – what does this mean? L419: The case has not been proven for this statement. L424: Difficulty cannot be "large" L437: I do not believe that this method could be used elsewhere. Most permafrost regions receive snow therefore negating or reducing its potential utility. L440: Misjudgements is not the correct term here. L445: Please revise the grammar in this sentence. L446: Please revise the grammar in this sentence. L454: "In compliance" is not used correctly. L462-463: I do not believe this study has adequately demonstrated this. Please remove.

References Riseborough, D.: The effect of transient conditions on an equilibrium permafrost-climate model, Permafrost and Periglacial Processes, 18(1), 21–32, doi:10.1002/ppp.579, 2007.

---

## Author Comment (AC1) · 28 May 2017

**Response to Referee #1**

We appreciate you very much for your comments concerning our manuscript entitled "A New Map of The Permafrost Distribution on The Tibetan Plateau" (MS No.: tc-2016-187). Those comments are valuable and helpful for improving our manuscript. We followed all comments and made revision and answers carefully. Revised portions are marked in red in the revised manuscript. The page, line, and figure numbers refer to our revised manuscript. And, a point-by-point reply to the comments are listed below.

**Main comments**

1. LST obviously is not equal to ground surface temperature (GST) that is required to drive TTOP. You acknowledge this in Section 3.1, however: (A) this needs to be much more explicit as you describe your methods. (B) Does it not make it extremely difficult to interpret values obtained under snowcover as snow surface temperatures are likely to be much lower than GST even in shallow (albeit likely cold and dry, therefore lower thermal conductivities than a temperate snowpack) snowpacks of the TP, especially on clear nights where high emissivities will cool the snow surface much more than the GST. I think it is really important you at the very least quantify how significant this problem will be in your study region and probably try to introduce a term that accounts for the offset between LST and surface temperatures under snow on the ground conditions. Additionally I think this statement is wrong:

   "In this study, the land surface temperature was directly used as the upper boundary conditions in the model; therefore, the LST calculation procedure with air temperature and n factor was omitted."

   LST is likely to be very similar to near surface air temperature – this approach therefore ignores the n-factors which are important in describing the offset between air and ground surface conditions particularly under snowcover as described above, also effects of vegetation (less significant perhaps).

Response:

It is a good question that also raised by another reviewer. We discussed this issue in a great detail within our research group. Ground Surface Temperature (GST) is defined as the surface or near-surface temperature of the ground (bedrock or surficial deposit), measured in the uppermost centimeters of the ground. The snow and vegetation might play significant influences on the relationship between the remote sensing LSTs and the GSTs, the influences is depending on the snow depth and duration (Zhang, 2005), vegetation height and coverage.

The spatial distribution of snow cover over the Tibetan Plateau (TP) varies quite greatly. The most wide-spread snow cover were found in southeastern part of the TP, and some alpine regions with the elevation higher than 6000 m (Qin et al., 2006; Pu et al., 2007); the snow cover is rare, shallow (< 3 cm) and existed in a short duration (mostly existed less than one day for one single snow event) due to very strong solar radiation and wind in the vast interior and the north of the TP, where the permafrost most developed, and

is our major study area in this manuscript (Che et al., 2008; Huang et al., 2017). The thin snow cover with short duration mainly has a cooling effect on LST due to the high surface albedo of fresh snow and a rapid process of snowmelt (Zhang, 2005). The duration of the cooling effect may be very short, thus it may have very little effect on the LST in average for certain period of time.

Generally, the soil surface beneath the vegetation layer have a higher temperature than the canopy surface, depending on the height and coverage. The alpine ecosystem in permafrost regions and its vicinity are all composed of grassland, characterized by dwarf and sparse plants. The vegetation coverage in most of the permafrost region was less than 30% (Lehnert et al., 2015), and even less than 10% in the middle and western TP.

In view of the actual condition of both snow cover and vegetation on the TP, there are only slight differences between MODIS LST and GST measured in meteorological stations in average, and the differences in thawing and freezing indices from both datasets were much small in our study area. In the revision of this manuscript, we added two figures of the snow depth (Fig.7, edited after Che et al.(2008)) and vegetation types (Fig.8, edited after Wang et al.(2016)) to show the possible regions influenced by the snow cover and vegetation.

In the revised version, we explained this in the *Section 4* (P.13 L.381-397).

[Figure]

**Figure 7.** Annual average snow-depth distributions on the Tibetan Plateau from 1979 to 2014 (edited after Che et al.(2008))

[Figure]

**Figure 8.** Vegetation types of the permafrost zone on the Tibetan Plateau (edited after Wang et al.(2016))

2. You don't give any evaluation of the MODIS product – how well does it perform in the region? What are the uncertainties under snowcover (wet/dry), vegetation, arid soils etc. You have AWS data from the permafrost field campaign you mention in Section 2.1 which may give you some clue if you measure surface temperatures (of coarse point/spatial scaling needs to be acknowledged). Cite the literature that has looked at uncertainties in MODIS LST regionally/globally and give some incites on how you expect this to affect your model setup.

Response:

Thanks for the comments and suggestions. The evaluation of the modified MODIS LST averages in the permafrost region on the TP has been done, which was described in detail in Zou et al. (2014). Briefly, the multiple linear regression model combined all four observations of Terra and Aqua MODIS LST were established to estimate the mean daily GST. The model validation in three permafrost sites (e.g. Xidatan, Wudaoliang and Tanggula) showed that the determination ($R^2$), mean error (ME), mean absolute error (MAE) and root mean squared error (RMSE) was 0.91 to 0.93, -0.21 to 1 ℃, 2.28 to 2.42 ℃ and 2.96 to 3.05 ℃, respectively (Zou et al., 2014). The uncertainties were mainly came from the following factors:1) temperature differences caused by the time offset between half-hour interval of ground-based observation and satellite overpass time; 2) the mismatch of spatial scales between point and pixel observation. Moreover, the three sites (with different vegetation types: including alpine meadow, alpine steppe and alpine desert) located at the continuous permafrost zone on the TP with altitude above 4000 m, the observed GST data could be used to calibrate the MODIS LST due to their high representativeness of climate condition in the permafrost region.

In the revised version, we have added the evaluation of MODIS LST briefly in the *Section 2.2.2* (P.6 L.189-191).

On the other hand, we cited some papers about the evaluation of global and regional MODIS LST products. The results of the radiance-based and temperature-based validation indicated that the accuracy of the global MODIS LST product is better than 1 ℃ in most cases, including lakes, homogeneous vegetation and soil sites in clear-sky conditions (Wan et al., 2002 and 2004; Coll et al., 2005; Wan and Li., 2008; Wan., 2008). In addition to validate the MODIS LST with the in-situ measurements at the same time within the footprint of the satellite sensor, some studies focused on the accuracy of LST averages for longer time periods, which is crucial for permafrost modelling. Langer et al. (2010) and Westermann et al. (2011) focused on weekly averages and demonstrated an agreement generally better than 2 ℃ for MODIS LST for the summer season, at permafrost sites in Siberia and Svalbard, respectively.

In the revised version, we cited the papers in the *Section 2.2.2* (P.6 L.164-169).

3. MODIS LST is a coarse resolution 1 km product. This of course will make any kind of discrimination of permafrost units at the subgrid scale difficult – mainly important on the north slope of Himalaya and other mountain regions of complex topography on the TP. It should be thoroughly discussed what limitations this poses for your results.

Response:

In our opinion, the changes of LST as well as air temperature with the horizontal distance are much greater in the north slope of Himalaya and Gangdis, where the slope is much steeper than that in the interior TP. It is said that the LST differences between both adjacent pixels are much larger, and is more sensitive to boundary identification. Moreover, there is almost no validated permafrost distribution data in these mountainous regions, due to the difficulties to carry out investigation.

In the revised version, we discussed the performance of TP-2016 in complex terrain in *Section 3.3* (P.10 L.294-298).

4. Bedrocks and debris slopes do not seem to be included in your "soil" classes and presumably are important land classes in your region. How do you deal with these?

Response:

In this study, the bedrocks and debris slopes were classified into Gelisols according to the soil classification criterion described in Li et al. (2015a and b). Owing to bedrocks and debris slopes are not much and generally located at much higher elevation than the lower limit of permafrost, mostly around the high mountain peaks (eg. The Kunlun, Tanggula, Himalayas and Gangdis mountains) where were glaciated and strongly weathered. It was said that even the simulation for such regions were not so accurate, however it does not affect the permafrost distribution modeling.

5. Why not present actual values derived from the TTOP model instead of just a binary map? This would be interesting to see eg. Where warm/cold permafrost exists.

Response:

Thanks for your excellent advice. We have change the binary map with the actual values derived from the TTOP model. In additional, the TTOP values were divided into six intervals so that the reviewers and potential readers can read and analyze conveniently.

The revised figure is as below, and could be found in the P.25.

[Figure]

**Figure 2.** Spatial distribution of permafrost with the derived TTOP on the Tibetan Plateau

6. How do you incorporate the effect of solar radiation (slope, aspect + possible horizon, sky view factor) into your five investigated regions (IR) that you use as validation? From reading section 2.1.2 it appears that you determine a lower limit of permafrost (LLPs) and extrapolate this across your IR. However this sentence:

   "The permafrost map was generated for each IR based on the criteria of LLPs in different conditions combined the digital elevation model (DEM) data…"

   Suggests you do something which may account for at least aspect. This needs to be well described as forms the basis of your evaluation. We need to know how well we can trust this and what uncertainties are involved in these validation datasets.

Response:

In the investigation, the influence of aspect on the LLP have been considered when the boreholes were set. The LLP was determined based on the linear regression relationship between MAGT and elevation of boreholes, and the elevation where MAGT equals to 0 ℃ was regarded as the LLP. In the permafrost mapping of each IR, the boreholes were classified into three types: north-facing, south-facing and east-west facing slopes and the LLP of each type was determined respectively; then, the permafrost distribution were generated based on the LLPs of different aspects and the digital elevation model (DEM) data, and a portion of the observed results of boreholes and geophysical methods (GPR and TEM) was reserved to validate the maps (Li et al., 2012; Zhang et al., 2012). For example, the results of GZ IR showed that the LLP was about 4950 m

for north-facing, 5000m for east-west-facing, and 5100 m for south-facing slopes (Chen et al., 2016).

In the revised version, we added this statement in the *Section 2.1.1* (P.4-5 L.130-138).

7.  In the validation exercise with the 2016 map we see large difference in kappa values: 0.38 – 0.78. Some discussion is given (p.14 l.439-448) which as far as I can tell indicates that smaller IR are more accurate due to density of measurements. However, best (0.78, WQ) and worst performing (0.38, AEJ) are roughly the same size. It is important you discuss these difference in performance with respect to how well you think your model performs in the various regions eg. Uncertainties such as complex topography, LST-GST offset etc.

Response:

The TTOP model identify the permafrost boundary with a temperature threshold, which is sensitive to the temperature differences in horizontal distance. As we described in the *Comments 3*, the TTOP model might perform better to identify the permafrost boundary in the regions with complex terrain because of sharp changes in LSTs within short distances, such as the WQ, B-Q and XKL IR. For GZ and AEJ IRs, where land surfaces are much flat, and so called lower surface relief, LSTs changes with distances are small, the TTOP model performs not as good as other IRs. For AEJ IR, the performance of TTOP model was worst, because of there is no soil pits in the investigation and the soil condition inferred completely from the relationship between the environmental factors and the soil samples of the other four IRs.

In the revised version, we added this statement in the *Section 3.3* (P.10 L.294-298).

8.  How long are your borehole records? These need to be better described in your data section. You show in Figure 3 how the new map better represent seasonally frozen ground eg. subset a3-c3. But how do we know whether the model is better or simply that the permafrost has thawed in this region over the past 20 years. The borehole measurements are not contemporary with the old permafrost maps as I understand – but that needs to be described as stated above. Additionally, in validation you compare a map derived from 2003-2012 MODIS data with borehole records of possibly another period. Basic point: it seems that comparability of different maps and validation datasets is problematic and these issues should be discussed well.

Response:

It is common that the ground temperatures of permafrost were increasing during the last several decades, but the increasing rates were much lower than that in circumpolar regions, and it was even much lower for warm permafrost (Wu and Liu, 2004; Wu and Zhang, 2008; Smith et al., 2005; Romanovsky et al., 2010; Smith et al., 2010; Zhao et al., 2010). In the other words, the monitored changes of permafrost temperature near the permafrost boundary on the TP was not so much, and there is no data showed that the permafrost in these regions have disappeared. For example, we have deployed two permafrost temperature monitoring sites since the 1970s: Liangdaohe (N31.82°,

E91.74 °), at the south permafrost margin area where the permafrost is sporadic with an area of less than 1 km$^2$ and its thickness is more than 60m, and Xidatan (N35.72 °, E94.09 °), the north margin and the thickness is about 20m. The permafrost at these two margin areas does not disappear until now. Furthermore, an ice-rich layer commonly exists at the top of a permafrost layer that the thawing process needs much more time and more energy input. Therefore, the changes of the permafrost distribution on the TP is limited in the past several decades and that is the basis of comparison between the boreholes investigation and different maps. Although the permafrost degradation was obvious, it mainly occurred as the temperature increasing of ground temperature and deepening of active layer, rather than the disappearance of permafrost.

In the revised version, we added this description briefly in the *Section 4* (P.14 L.440-444).

9. In Section 3.5 you compare maps 1996, 2006 and 2016. What is the main message from this comparison? How do you disentangle changes in permafrost distribution computed from possible actual changes in MAAT over the last 20 years and differences due to different methods and sources of uncertainty? It would be good to be clearer about what the various differences that are observed are correlate with i.e. complex topography, latitude, data scarcity etc.

Response:

The purpose of the comparison was to show that the result in this study was the more accurate map of the permafrost distribution in the contemporary climate. There is a quite small change in permafrost distribution on the TP over the past several decades as we described in *Comment 8*. Therefore, the primary causes for differences in the three maps were the different methods and data sources (data quantity, accuracy, and regional representativeness). The difference areas are mainly distributed in the periphery of the continuous permafrost and high mountainous regions where local factors play more important roles. In view of the TP-2016 has a better performance in complex terrain, it is reasonable to infer that the accuracy of the TP-2016 is higher than that of the other two maps in the difference areas. In addition, the changes in the permafrost distribution should be much smaller than the difference caused by the different methods and data source.

In the revised version, we discussed this issue in the *Section 4* (P.14 L.440-445).

10. Issue of permafrost conditions out of equilibrium with today's climate i.e. warming permafrost conditions, should be discussed. Surface forcing could indicate no permafrost according to today's conditions – but there exists a long response time of permafrost bodies to modern atmospheric conditions. Therefore any map based on a contemporary forcing likely underestimates permafrost extent and especially, arguably the most interesting/disruptive warming/thawing permafrost bodies. This fact does not have an easy solution, but certainly should be discussed.

Response:

It is really a good question. Permafrost on the TP is out of equilibrium under global climate warming. As the reviewer said, there exists a long response time of permafrost bodies to atmospheric conditions, even millions of years. All the maps using modern climate conditions are difficult to solve the problem because the evolution of permafrost is a large time scale issue. In the view of the solution of permafrost identify, the boundary of permafrost is the most sensitive region to the climate change and has close relation with the contemporary climate. Therefore, the essential of modern permafrost mapping is how to improve the accuracy of surface forcing and the soil parameters to identify the boundary. In this study, the TTOP modelling based on remote sensing LST and plenty of current in situ soil parameters observation shows a high accuracy in the validation. However, the results might not capture thawing permafrost bodies, and more works still were needed to be done.

In the revised version, we discussed this issue in the *Section 4* (P.14 L.436-447).

11. How do you identify "thawing regions" (Section 5 l.460 and mentioned throughout text). This would require some form of transient modelling that demonstrates a transition from permafrost to non-permafrost conditions? As far as I can tell you are equating detection of seasonally frozen ground to thawing conditions. If you use "seasonally frozen ground" in figures also use this in text, otherwise confusing to reader that likely associates the word "thaw" with a change in permafrost conditions.

Response:

To avoid any confusions, all the "thawing region" was instead of "seasonally frozen ground" in the revised manuscript. Thanks for pointing out.

12. Language of Section 4 is very poor in sharp contrast to rest of paper which is generally fairly good.

Response:

The language of *Section 4* has been polished carefully.

**Minor comments**

1. Might be worth citing Gruber 2012 (cited later in paper) in your introduction where you discuss TP permafrost maps.

Response:

The reference of *Gruber (2012)* has cited in the *Introduction* and removed from the *Discussion* section (P.2 L.61-64).

2. p.6 l.174: massive -> numerous

Response:

The "massive" has revised to "numerous" (P.6 L.174).

3. p.6 l.175: describe what HANTS is and why you use it. Details can be left to the reference but reader needs to know the basic purpose of this method.

Response:

The description of HANTS has been added and the details was left to the reference (P.6 L.176-179).

4. p.6 l.183: what are MODIS overpass times? how many Swathes used?

Response:

The overpass times and the total swathes of MODIS have been added (P.6 L.180-183).

5. p.7 eq.4: mention that kt/kf comes from properties derived in Section 2.2.3. Make this link more obvious in text.

Response:

The mention that $k_t/k_f$ comes from properties derived in Section 2.2.3 has been added (P.8 L.230).

6. p.9 l.260: "decreases gradient" -> "decreases linearly", is that what you mean?

Response:

That is exactly what we mean. We have revised accordingly (P.9 L.261).

7. p.13 l.415: I would rather say medium spatio-temporal resolution. I don't think 4 daily values at 1 km qualifies as "high res" on either dimension.

Response:

Thanks for the comments. The "high" has changed to "medium" (P.14 L.416).

8. p.13 l.416: "it can reflects" -> "it can represent"

Response:

We have revised accordingly (P.14 L.417).

9. p.14 l.421-422: "The improvement of upper boundary conditions of permafrost model and the employed of massive reliable in situ observed datasets make the high modelling accuracy achieved." -> "The improvement of upper boundary conditions of the permafrost model and the use of large quantities of reliable in situ observed datasets, leads to a high modelling accuracy."

Response:

Thanks. Changed accordingly (P.14 L.421-422).

10. Based on comment above about comparability of maps and validation data, I don't think this statement is so straightforward (p14 l439): "Although TP-2016 performed better than TP-1996 and TP-2006 and showed substantial agreement with the investigated results, it still results in some misjudgments".

Response:

To avoid any confusions, the sentence has removed.

11. What is the permafrost distribution of figure 1 based on ?

Response:

The permafrost distribution of figure 1 is the result in this study without the unfrozen ground that we want to show the IRs are right on the boundary. To avoid any confusion, it has been changed to the permafrost map made in 1996 in the revised version (P.24).

12. Acknowledgments: remove "Level 1 and".

Response:

The "Level 1 and" has removed from the Acknowledgments. Thanks.

References:

[revised manuscript text omitted]

---

## Author Comment (AC2) · 28 May 2017

**Response to Referee #2**

Thank you very much for your comments concerning our manuscript entitled "A New Map of The Permafrost Distribution on The Tibetan Plateau" (MS No.: tc-2016-187). Those comments are valuable and helpful for improving our manuscript. We followed all comments and made revision and answers carefully. The changes are marked in red in the revised manuscript. The page, line, and figure numbers refer to our revised manuscript. And, a point-by-point reply to the comments are as following:

1. MODIS LST. The authors use MODIS LSTs as the key input for their model of permafrost distribution. However, MODIS LSTs measure a combination of different surfaces including the snow surface. If, as the authors postulate, there is only minimal snow cover across the region and correspondingly that MODIS LSTs can be used in the winter then this is all fine. However, the authors have not shown conclusively that snow cover impacts on LST retrievals can be ignored for their region. Addressing this point is a necessity for this manuscript to be considered suitable for publication in The Cryosphere. Likewise, there is certainly some effect of canopy cover in the summer which has been ignored by the authors. It would be useful if the authors examined the ecotype related impacts on the LST and correspondingly how this may affect the distribution of permafrost in the region. I also suggest that the authors produce an additional figure which shows a 1st panel with the estimated regional snow depth across the area (either from reanalysis or other datasets) and a 2nd panel that shows the spatial distribution of vegetation classes (broadly) across the region so that as reviewers we can determine the degree to which this issue may be problematic. Another issue with the MODIS LST that I find concerning is that the authors make the claim that MODIS is preferable to interpolation for temperature (it seems to be in the context of air temperature). A number of studies have found issues with MODIS-derived (or aided) air temperature products with only minimal improvements being observed (if at all) in terms of cross-validation.

   Although I do think that MODIS products have utility for permafrost purposes, more work must be done to demonstrate that these products offer improvement over high resolution interpolation of station-based temperature products. It is important that the permafrost community ensures that the usage of LSTs from MODIS for driving permafrost models is assessed at each usage given the spatial heterogeneity of the factors influencing MODIS LSTs.

Response:

It is a very good question that also raised by another reviewer. The snow and vegetation might play significant influences on the relationship between the remote sensing LST and the ground surface temperature (GST), the influences is depending on the snow depth and duration (Zhang, 2005), vegetation height and coverage. On the Tibetan Plateau (TP), the most wide-spread snow cover were found in the southeastern region and some alpine regions with the elevation higher than 6000 m (Qin et al., 2006; Pu et

al., 2007). In the vast interior and northern TP, where the permafrost most developed and is our major study area in this manuscript, the snow cover is rare, shallow (< 3 cm) and existed in a short duration, mostly existed in several hours for one single snow event, due to very strong solar radiation and wind (Che et al., 2008; Huang et al., 2017). In this case, the thin snow cover with short duration mainly has a cooling effect on LST due to the high surface albedo of fresh snow and a rapid process of snowmelt. The duration of the cooling effect may be very short, thus it may have very little effect on the LST in average for certain period of time (Zhang, 2005). Generally, the soil surface beneath the vegetation layer have a higher temperature than the canopy surface, depending on the height and coverage. The alpine ecosystem in permafrost regions and its vicinity are all composed of grassland, characterized by dwarf and sparse plants. The vegetation coverage in most of the permafrost region was less than 30% (Lehnert et al., 2015), and even less than 10% in the middle and western TP. Therefore, there are only slight differences between MODIS LST and GST, and even smaller in the thawing and freezing indices in our study area. In the revised version, we added two figures of the snow depth (Fig.7, edited after Che et al.(2008)) and vegetation types (Fig.8, edited after Wang et al.(2016)) to show the possible regions influenced by the snow cover and vegetation. The two figures were as follow and could be found in the revised manuscript (P.30-31).

In the revised version, we explained this in the *Section 4* (P.13 L.381-397).

[Figure]

**Figure 7.** Annual average snow-depth distributions on the Tibetan Plateau from 1979 to 2014 (edited after Che et al.(2008))

[Figure]

**Figure 8.** Vegetation types of the permafrost zone on the Tibetan Plateau (edited after Wang et al.(2016))

We are sorry about the inappropriate wording of "preferable". Actually, what we want to say is the remote sensing products are available data source besides the temperature interpolation and reanalysis datasets, especially for the remote areas with limited observations such as the Tibetan Plateau. As we discussed in the paper, the meteorological sites on the Tibetan Plateau are scarce and unevenly distributed (more in the Eastern TP and less in the Western TP; more in lower elevation and less in higher elevation), which lead to the high uncertainty of temperature interpolation in this region (Lin et al., 2002; Li et al., 2003). There is almost none meteorological sites in the permafrost region of the western TP, where permafrost most developed region, and this is why we trying MODIS LST products. In this study, all four observations of Terra and Aqua MODIS LST were employed to establish the multiple linear regression models of the daily mean GST estimation. The models validation in three permafrost sites showed that the determination ($R^2$), mean error (ME), mean absolute error (MAE), and root mean squared error (RMSE) of 0.91 to 0.93, -0.21 to 1 ℃, 2.28 to 2.42 ℃, and 2.96 to 3.05 ℃, respectively (Zou et al., 2014). The uncertainties were mainly due to the several factors, including temperature differences caused by the time offset between half-hour interval of ground-based observation and satellite overpass time, and the mismatch of spatial scales between point and pixel observation. Although three sites, located at the continuous permafrost and underlie different vegetation types, seems few in validation and calibration, they will make the modified MODIS LST much be close to the real values.

In the revised version, we rewritten the earlier sentence with "preferable" as below: **And they were used as effective alternatives for LST, especially for in-situ observation limited regions, such as the TP (Zhang et al., 2004)**. Then, we added the evaluation of MODIS LST in the *Section 2.2.2* (P.6 L.189-191).

2. TTOP modelling output. The others provide a simple binary term for the presence

or absence of permafrost that is useful in the context of total are numbers but also means that huge amounts of information are unavailable. A map of TTOP temperatures could be useful in interpreting areas most susceptible to future change and also for the purposes of understanding permafrost thicknesses under a variety of environments. I would highly recommend the authors at least present one map showing the spatial distribution of TTOP temperatures.

Response:

Thanks for your excellent advice. We have change the binary map with the actual values derived from the TTOP model according to the permafrost distribution. In addition, the TTOP values were divided into six equal intervals so that the reviewers and potential readers can read and analyze conveniently.

The revised figure is as below, and could be found in the *P.25*.

[Figure]

**Figure 3.** Spatial distribution of permafrost with the derived TTOP on the Tibetan Plateau

3. Uncertainties. Given the uncertainties that may be present in the LST products and in distributing rk across the landscape, it would seem important that some assessment of uncertainty is provided for the estimates of total permafrost area. It also may be a little optimistic to assume that all glacier area would correspond to permafrost area given the vast range of climates in the region. Such an assumption would require a very detailed assessment to rationalize – I'd prefer it be left out.

Response:

The uncertainty analysis was conducted in R statistical software (version 3.3.1, www.r-project.org) using the Percentile Method combined the uncertainties of MODIS LST and in distributing $r_k$ across the soil types, and we use a 90% confidence interval to find the range of total permafrost areas. The results showed that the median permafrost area was $1.06 \times 10^6$ km$^2$, with 90% confidence interval of $0.97 \times 10^6$ km$^2$-$1.15 \times 10^6$ km$^2$.

To avoid any confusions, the assumption that all glaciers area correspond to permafrost area has been left out in the revised version.

4. Non-equilibrium permafrost. The authors should consider the results of Riseborough (2007) when evaluating their TTOP model output and particularly in the context of non-equilibrium permafrost. Is the region warming and if so would this be impacting the distribution of permafrost as measured from this equilibrium model? One of the challenges in using a MODIS derived product is that the relatively short period of coverage makes it more challenging to model in hindcast.

Response:

It is really a good question. That is very interesting paper that we have read it seriously. In fact, with changing climate, short-term energy imbalances between the active layer and permafrost result in transient departures from the equilibrium condition. The TTOP model has an error in the top-of-permafrost temperature obtained with the equilibrium model that is higher where permafrost temperature is close to 0 °C. Permafrost on the TP is out of equilibrium under global climate warming, and there exists a long response time of permafrost bodies to atmospheric conditions. Any map based on a contemporary forcing likely underestimates permafrost extent. In the view of the solution of permafrost identify, the boundary of permafrost is the most sensitive region to the climate change and has close relation with the contemporary climate. Therefore, the essential of modern permafrost mapping is how to improve the accuracy of surface forcing and the soil parameters to identify the boundary. This study aims to improve the surface forcing with MODIS LST although relatively short period of coverage. The TTOP modeling in this study based on remote sensing LST and current soil parameters shows a high accuracy in the validation. However, the results may cannot capture thawing permafrost bodies, and more works still were needed to be done.

In the revised version, we discussed this issue in the *Section 4* (P.14 L.436-447).

Minor points:

L16: Remove "mostly"

Response:

Removed accordingly.

L27-28: Identifying 'thawing regions' seems unclear to me.

Response:

The "thawing regions" should be seasonally frozen ground, we have removed the statement of "thawing regions" in the sentence. To avoid any confusions, all the "thawing region" was instead of "seasonally frozen ground" in the revised manuscript. Thanks for pointing out.

L41-42: Urgent is perhaps a bit strong of a word here, as is 'situation'.

Response:

The sentence "**Therefore, understanding the current permafrost situation on TP has become particularly urgent**" is a repetitive statement of the earlier sentence in the manuscript, which has removed and we have revised the earlier sentence as "**Moreover, an accurate contemporary permafrost distribution map is of significant importance for serve as a standard to estimate permafrost degradation and as a basis for further quantitative research.**" (L.40-41).

L46: "there is great variation" -> "there is considerable variation"

Response:

Revised accordingly (L.45). Thanks.

L49-50: This sentence should be re-written to be clearer. At present, it makes no sense.

Response:

It is a repetitive sentence and has been removed.

L51-52: What is the difference between a topographic map and a base map?

Response:

There is no difference between a topographic map and a base map, the sentence has revised as "**In 1980s and 90s, permafrost maps were compiled by different scientists, and the permafrost boundaries were plot on topographic maps by hands with conventional cartographic techniques**" (L.49-50).

L54: "On the" -> "On"

Response:

Revised accordingly. Thanks.

L55-56: This statement is not true. GIS techniques were used before 2000…

Response:

The sentence has revised as "**After 2000, GIS software began to be applied to the permafrost mapping on the TP**" (L.53-54).

L58: What does "stability of elevation" mean?

Response:

The "stability of elevation" means that elevation changes little or even remain constant for a long period of time. To avoid any confusion, the sentence has removed.

L74-75: I do not agree with this sentence. Temperature and reanalysis data have a higher

temporal resolution than MODIS and can be interpolated more accurately. In my experience, MODIS LST products in the Subarctic and Arctic are not suitable alone for characterizing spatial variations in temperature.

Response:

We agree the reviewer's comments about the issue of the applicability of MODIS LST. Actually, what we want to say is the remote sensing products are other available data source besides the temperature interpolation and reanalysis datasets, this is very important data especially for the remote areas with limited observations such as the Tibetan Plateau.

We have changed the sentence as "**And they were used as effective alternatives for LST, especially for in-situ observation limited regions, such as the TP (Zhang et al., 2004).**" (L.72-73).

L80: I agree. The authors should provide examples of this validation.

Response:

The sentence make no sense here, we have removed it and added some examples of MODIS LST validation in *L.164-169* and the validation in this study in *L.189-191*.

L87: Remove "plenty of"

Response:

Removed accordingly.

L89: Remove "perfect"

Response:

Removed accordingly.

L95: Remove this sentence

Response:

Removed accordingly.

L96: Remove "combined"

Response:

Removed accordingly.

L113: What is "drilling method". The grammar seems a bit off.

Response:

The "drilling method" has change to "borehole drilling" (L.110).

L136-137: The grammar in this sentence should be revised.

Response:

The sentence was re-written as: **The lower limit of permafrost (LLP) was determined based on the linear regression relationship between MAGT and elevation of boreholes, and the elevation where MAGT equals to 0 ℃ was regarded as the LLP (Li et al., 2012; Zhang et al., 2012; Chen et al., 2016)** (L.130-132).

L157: "mostly widely" -> "most widely"

Response:

Revised accordingly (L.152).

L174: "massive missing values" -> "many missing values"

Response:

The "massive missing values" has revised to "numerous missing values" (L.174).

L175: "Harmonic ANalysis Time" -> "Harmonic Analysis Time"

Response:

Revised accordingly (L.175).

L176-177: Remove sentence or combine with earlier sentence

Response:

The sentence has combined with the earlier sentence (L.175-178).

L197: What is "stability of the data"?

Response:

The "stability of the data" has removed to avoid the confusion.

L197: I prefer FDD and TDD or sFDD and sTDD to DDF and DDT.

Response:

The DDF and DDT has changed with FDD and TDD in the revised version.

L199: Amend to: "Soil thermal characteristics were modeled according to parameters measured from soil types encountered in the field".

Response:

Revised accordingly (L.201). Thanks.

L232: We do not need a sentence to tell us that an abbreviation was used.

Response:

The sentence has removed.

L259-260: Amend: "increases with increasing" and "decreases…decreases".

Response:

Revised to "**The permafrost continuity decreases linearly as the elevation decreases and the ground temperature increases with increasing distance from the central region.**" (L.261-262).

L264: This sentence could be shortened with the use of brackets.

Response:

The sentence has been shortened with use of brackets as "**Some unfrozen ground exists in the southeast margin of the TP, whose area is approximately 0.03×10$^6$ km$^2$ (account for 1% of the total TP area)**" (L.266-267).

L280: Boreholes are not "convincing evidence" of permafrost rather they can determine if permafrost exists or not. This sentence should be revised.

Response:

The sentence has revised as "**The boreholes can determine whether permafrost exists or not**" (L.269).

L311: "…correct…correct" – please revise.

Response:

The sentence has revised as "**Although the correct pixels were few, the locations in the eastern part were just at the geographic boundary of permafrost**" (L.304-305).

L329: "overcomes this shortcoming" – That is not necessarily proven in this study.

Response:

The statement of "overcomes this shortcoming" is inappropriate, we have removed it in the sentence.

L360: "lower distance difference" – Please clarify.

Response:

The "lower distance difference" means the widths of seasonally frozen ground along the highway transect identified from TP-2016 have smaller difference with the investigation results than that of both TP-1996 and TP-2006. We have revised the

"lower distance difference" to "**smaller width difference**" to keep the agreement with other sentences.

L389-392: This sentence is confusing – please revise.

Response:

We have re-written the sentences as "**The dataset used in the earliest maps (compiled in 1980s and 90s) including air temperature, field data, aerial photographs, satellite images and many relevant maps (Tong and Li, 1983; Shi and Mi, 1988; Li and Cheng, 1996).**" (L.398-399)

L399: "are unevenly" -> "unevenly"

Response:

Revised accordingly.

L400: What is poor representativeness?

Response:

The "poor representativeness" means the monitoring sites is very few in the permafrost region of TP that the extrapolated air temperature may not reflect the actual climate condition of permafrost region. The sentence has revised as below: **In addition, high uncertainty exists in the air temperature interpolation because of the scarcity, unevenly distributed (more in the Eastern TP and less in the Western TP; more in lower elevation and less in higher elevation; very few in permafrost region) monitoring sites, resulting in the low accuracy of extrapolated air temperature of the TP (Lin et al., 2002; Li et al., 2003), especially for the permafrost region** (L.404-407).

L404: "the most accurate" – Remove this sentence.

Response:

The sentence has removed.

L409: Poor sentence grammar – Please revise.

Response:

The sentence has removed. And, the reference *Gruber.(2012)* was cited in the *Introduction* according to the other reviewer's suggestion (L.61-64).

L416: "reflects" -> "reflect"

Response:

The "reflects" has changed to "represent" (L.417).

L418: "high representativeness" – what does this mean?

Response:

The "high representativeness" means the temperature observed by automatic weather stations in typical permafrost region can represent the actual climate condition. In view of the wording "representativeness" may not a common word, we have revised the sentence as below: **In addition, the MODIS LST data was calibrated by the ground-based LST observations obtained from automatic weather stations in typical permafrost regions (Zou et al., 2014), which is corresponding to actual climate condition of permafrost region** (L.418-420).

L419: The case has not been proven for this statement.

Response:

We have revised the sentence as below: **In addition, the MODIS LST data was calibrated by the ground-based LST observations obtained from automatic weather stations in typical permafrost regions (Zou et al., 2014), which is corresponding to actual climate condition of permafrost region**(L.418-420).

L424: Difficult cannot be "large"

Response:

The sentence was revised as: **In the earliest maps, only some observed data from the field sites along Qinghai–Xizang Highway were used for map evaluation (Tong and Li, 1983; Shi and Mi, 1988; Li and Cheng, 1996)** (423-424).

L437: I do not believe that this method could be used elsewhere. Most permafrost regions receive snow therefore negating or reducing its potential utility.

Response:

The statement of "which could be used for other places" has removed.

L440: Misjudgements is not the correct term here.

L445: Please revise the grammar in this sentence.

L446: Please revise the grammar in this sentence.

Response:

The sentences of *L440, L445, L446* were re-written in the revised version.

L454: "In compliance" is not used correctly.

Response:

The "In compliance with" has revised to "**From the validation with**" (L.453).

L462-463: I do not believe this study has adequately demonstrated this. Please remove.

Response:

The sentence has removed.

References:

[revised manuscript text omitted]

---

## Author Response (AR2)

**Response to Referee #1 (Report #2)**

We appreciate your comments and suggestions concerning our manuscript entitled "A New Map of the Permafrost Distribution on the Tibetan Plateau" (MS No.: tc-2016-187). They are valuable and helpful for improving our manuscript. We followed all comments and made revision carefully. Revised portions are marked in red in the revised manuscript. The page, line, and figure numbers refer to our revised manuscript. A point-by-point reply to the comments are listed below.

1.  I think it is important to differentiate between LST, which is a term that originates in the remote sensing community, eg. https://link.springer.com/referenceworkentry/10.1007%2F978-0-387-36699-9_79#CR866 and GST which is commonly used in the permafrost community. I know the authors are well aware of this differentiation and have published extensively on these topics as well as in their response to my original comment 1. I still do think it's important that the terms are applied consistently and rigorously in the manuscript to avoid confusion. For example in the Introduction at paragraph starting l.66 I feel like you should talk about GST observations, not LST. We typically measure GST on the ground, not LST (with exception of radiometers used for snow surface temp measurements, although these are calibrated for snow properties so the reading they give in absence of snow can be quite inaccurate) and in the original formulation of TTOP (Smith and Riseborough 1996) N factors are used to transfer air temperature to ground surface temperature NOT land surface temperature (what a satellites sees). N factors are used precisely to adjust for the effects of surface cover (veg/snow) which is not required for LST because LST measures the highest most layer ("skin-temp") of surface cover, whatever that happens to be at a given point in time. By definition, offsets generated by snow or vegetation layers are not relevant to LST.

    You discuss this in your discussion but I think it would be much more useful to lay down the theoretical framework in your intro (as you do in part) and make sure LST and GST are well defined and not used interchangeably and any justification you have for using LST as GST is offered early on (probably best in Methods) so the reader is not left confused about what you do.

Response:

Thanks for your comments and suggestions.

**Firstly,** we added the definitions of both "GST" and "LST" to differentiate them and avoid any confusion as below.

Definition of "GST" (P.3 L.67-69):

"… ground surface temperature (GST, defined as the surface or near-surface temperature of the ground (bedrock or surficial deposit) and measured in the uppermost

centimeters of the ground), …".

Definition of "LST" (P.3 L.72-74):

"… land surface temperature (LST, defined as the average temperature of an element of the exact surface of the Earth (e.g. surface of ground, vegetation canopy or snowcover) calculated from measured radiance; Gillespie., 2014) …".

**Then,** we checked the term "GST" and "LST" in the whole manuscript to make sure that they were used appropriately in the given context (e.g. the term in L.66, as you decribed in the comment, should be GST not LST). We marked the changes in red in the revised version.

**Finally,** the justification we have for using LST as GST is offered in *Methods* (P.8 L.229-233).

2. I still find structure missing in methods section particularly in terms of separating validation/driving data and the setup of the modelling scheme. I think a simple figure describing the modelling scheme would be a nice addition for the reader.

Response:

Thanks for the suggestion.

In the revised version, we added a simple flow diagram to describe the modelling scheme in *Section 2.3*, which divides the driving and validation data clearly.

[Figure]

**Figure 2**. Flow diagram of the modelling scheme

3. Gap filling has bias? As I understand HANTS (I'm using this source: https://mabouali.wordpress.com/projects/harmonic-analysis-of-time-series-hants/) it is a gap filling (among other functions) algorithm based on a kind of Fourier transform. My concern is that your MODIS LST obs are biased to clear sky

conditions, being an optical sensor. Gap filling using such methods cannot recreate part of the data distribution that doesn't exist in the available dataset (i.e. cloudy sky conditions). Admittedly we talk about arid/semi-arid TP were clear-sky conditions dominate. I would argue that theoretically, this approach is not ideal and difficult to transfer to more humid regions, but practically it's possibly OK for your area. I think this issue should be discussed though.

Response:

Thanks for your suggestion. As the reviewer suggested, we have checked the website and found that download website of the HANTS software is same with that of our manuscript.

In the revised version, we discussed the HANTS performance on the TP in the *Section 4* (P.13 L.400-403) as below:

"In addition, the HANTS algorithm might caused some bias under the cloudy sky condition, and therefore, further evaluation of the algorithm was not performed in this study because it has been proved to be an effective approach for filling the gap of MODIS LST data on the TP where clear-sky conditions dominated (Xu et al., 2013)."

Response:

Thanks for the suggestion. We have added *Figure 8* to show the snow depths and ecotypes in the revised version.

[Figure]

**Figure 8**. Annual average snow-depth (a; edited after Che et al., 2008) and vegetation types of the permafrost region (b; edited after Wang et al., 2016) on the Tibetan Plateau

4. Another point that I believe should be highlighted is that this representation may be useful for mapping purposes but the approach should not be used for modelling transient responses (i.e. climate change prediction). This future projections should use one-dimensional numerical modelling (e.g. NEST, GIPL model, or T-ONE).

Response:

Thanks for your suggestion, we have discussed this issue in the *Section 4* (P.14 L.441-444) as below:

"It is worth mentioning that the approach in this study is useful for mapping purposes, but it should not be used for modelling transient responses of permafrost to climate change which should use one-dimensional numerical models (e.g. NEST, GIPL model, or T-ONE).".

5. An additional consideration is the authors should add in a map showing the position of the study area at a broader scale. This could be combined with Figure 1.

Response:

We have modified the *Figure 1* to show the positions of the five investigated regions on a broader scale in the revised version.

[Figure]

**Figure 1**. Spatial distribution of the field survey regions on the Tibetan Plateau (based on the permafrost distribution map made in 1996)

6. Figure 6: Seems to be that the quality of this figure is not particularly high - would recommend submitting a higher resolution version.

Response:

Thanks. We have changed this figure to a higher resolution (600 dpi; Figure 7).

[revised manuscript text omitted]

---

## Author Response (AR3)

**Response to Editor**

We appreciate your comments and suggestions concerning our manuscript entitled "A New Map of the Permafrost Distribution on the Tibetan Plateau" (MS No.: tc-2016-187). We followed all comments and made revision carefully. Revised portions are marked in red in the revised manuscript. The page, line, and figure numbers refer to our revised manuscript. A point-by-point reply to the comments are listed below.

1. I know that it can be quite hard for non-natives to write in proper English. However, it needs to be free of mistakes and should be understandable. There are still some issues and I do not see in the track changes document that you made an effort to improve. In case you cannot find a person who can carefully proofread the manuscript, please get in touch with the Copernicus publisher. Some language editing is included in the page charges.

Response:

Thanks for your suggestion. We have improved the language of our manuscript in *LetPub Company* to ensure the paper quality.

2. L. 28 omit "much" and write "information" instead of "maps".

Response:

Thanks. Changed accordingly.

3. The new information about the climate is not understandable and needs to be slightly extended. A lower MAAT rage from -9.7 to 6.8 ℃ is large and should be put into regional context and context of elevation.

Response:

Thanks. The MAAT is spatially variable over the TP. We consulted some references for the data and used the "permafrost region" to limit the MAAT range. In the revised version, we amended the sentence in the *Section 1* (P.2 L.27-31):

"Due to its unique and extremely high altitude (mean elevation over 4000 m) (Qiu, 2008) and low mean annual air temperatures (generally lower than -2 ℃ with an intra-annual amplitude over 20 ℃ in the permafrost region) (Zhou et al., 2000; Yang et al., 2010), the Tibetan Plateau (TP) (Zhang et al., 2002; Zhang et al., 2014) possesses the largest areas of permafrost in the mid- and low-latitude regions of the world (Zhao et al., 2004; Zhao et al., 2010)."

4. The newly included sentence (P. 14) is basically a copy and paste from the review without own reflection. One could also use higher dimensional modelling. In addition, the models where not mentioned at any other place in the manuscript, I case you want to keep provide a reference.

Response:

Thanks for the comment. We did not give more information about other models because

they are not mentioned in the manuscript, and thus the discussion of them seems to be not appropriate. Therefore, to avoid any confusion, we only emphasize the limitation of our approach and omit the discussion about the other models. The sentence was revised as below (P.14 L.443-445):

[revised manuscript text omitted]

---

## Author Response (AR4)

**Response to Editor**

We appreciate your comments concerning our manuscript entitled "A New Map of the Permafrost Distribution on the Tibetan Plateau" (MS No.: tc-2016-187). We followed the comments and made revision carefully. Revised portions are marked in red in the revised manuscript. The page, line, and figure numbers refer to our revised manuscript. A point-by-point reply to the comments are listed below.

1. The new sentence about the temperature and permafrost is difficult to read as it is separated by several references. Please cite all references at the end of the sentence and reduce them to 4 (or max. 5). E.g. Qui (2008) is not needed here. Make sure the newly included references are also in the reference list (I did not check).

Response:

Sorry for the confusing sentence. Following the Editor's suggestion, we removed the redundant references (e.g. Qiu, 2008; Zhang et al., 2002, 2014) and revised the sentence as follows (P.2 L.27-30):

"Due to its unique and extremely high altitude (mean elevation over 4000 m) and low mean annual air temperatures (generally lower than -2 ℃ with an intra-annual amplitude over 20 ℃ in the permafrost region), the Tibetan Plateau (TP) possesses the largest areas of permafrost in the mid- and low-latitude regions of the world (Zhou et al., 2000; Zhao et al., 2004; Yang et al., 2010; Zhao et al., 2010)."

In addition, all the citations in the manuscript are listed in the *Reference Section*. Thank you.

[revised manuscript text omitted]